# Emergency medical service interventions and experiences during pandemics: A scoping review

Despina Laparidou[1], Ffion Curtis[2], Nimali Wijegoonewardene[1,3], Joseph Akanuwe[1‡], Dedunu Dias Weligamage[1,3‡], Prasanna Dinesh Koggalage[1,3‡], Aloysius Niroshan Siriwardena[1‡]*

1 Community and Health Research Unit, School of Health and Social Care, University of Lincoln, Brayford Pool, Lincoln, United Kingdom, 2 Department of Health Data Science, Liverpool Reviews & Implementation Group (LRiG), Institute of Population Health, University of Liverpool, Liverpool, United Kingdom, 3 Ministry of Health, Colombo, Sri Lanka

☯ These authors contributed equally to this work.
‡ JA, DDW, PDK and ANS also contributed equally to this work.
* nsiriwardena@lincoln.ac.uk

**Data Availability Statement:** Data are provided within the paper and its Supporting Information files together with additional data extraction files available from https://repository.lincoln.ac.uk/

## Abstract

### Background

The global impact of COVID-19 has been profound, with efforts to manage and contain the virus placing increased pressure on healthcare systems and Emergency Medical Services (EMS) in particular. There has been no previous review of studies investigating EMS interventions or experiences during pandemics. The aim of this scoping review was to identify and present published quantitative and qualitative evidence of EMS pandemic interventions, and how this translates into practice.

### Methods

Six electronic databases were searched from inception to July 2022, supplemented with internet searches and forward and backward citation tracking from included studies and review articles. A narrative synthesis of all eligible quantitative studies was performed and structured around the aims, key findings, as well as intervention type and content, where appropriate. Data from the qualitative studies were also synthesised narratively and presented thematically, according to their main aims and key findings.

### Results

The search strategy identified a total of 22,599 citations and after removing duplicates and excluding citations based on title and abstract, and full text screening, 90 studies were included. The quantitative narrative synthesis included seven overarching themes, describing EMS pandemic preparedness plans and interventions implemented in response to pandemics. The qualitative data synthesis included five themes, detailing the EMS workers' experiences of providing care during pandemics, their needs and their suggestions for best practices moving forward.

articles/dataset/EMS_pandemics_Repository_
Data_19_02_2024_xlsx/25245448.

**Funding:** The authors received no specific funding
for this work.

**Competing interests:** The authors have declared
that no competing interests exist.

## Conclusions

Despite concerns for their own and their families' safety and the many challenges they are faced with, especially their knowledge, training, lack of appropriate Personal Protective Equipment (PPE) and constant protocol changes, EMS personnel were willing and prepared to report for duty during pandemics. Participants also made recommendations for future outbreak response, which should be taken into consideration in order for EMS to cope with the current pandemic and to better prepare to respond to any future ones.

## Trial registration

The review protocol was registered with the Open Science Framework (osf.io/2pcy7).

## Background

Coronavirus disease 2019 (COVID-19), caused by severe acute respiratory syndrome coronavirus-2 (SARS-CoV-2), was first discovered in humans in Autumn 2019. By February 2020 it had spread around the world and was declared to be a global pandemic on March the 11[th] by the World Health Organization (WHO). The global impact of COVID-19 has been profound and attempts to manage and contain the virus have placed increased pressure on healthcare systems and Emergency Medical Services (EMS) in particular. Lessons learnt from previous pandemics, such as Severe Acute Respiratory Syndrome (SARS-CoV) in 2002, as well as the current COVID-19 pandemic, are essential for informing global pandemic preparedness plans [1], which are key tools for fighting the current and any future pandemics as well.

Emergency Medical Services, in particular, have a vital role within emergency preparedness systems as they are on the front line of responses to the urgent medical needs of patients. There have been few studies investigating EMS' role in pandemic preparedness prior to the COVID-19 pandemic, but there has been a flurry of evidence since the current pandemic started. For example, one study [2] investigated EMS personnel's willingness to respond to an influenza outbreak and found that those workers who were concerned, but confident in their abilities and knowledge in flu as well as in workplace safety, were more likely to report for duty. A more recent study [3], aiming to assess EMS' available resources, including personal protective equipment (PPE) availability, and institutional policies and practices during the COVID-19 pandemic found there was a need for better education and training around clinical symptom recognition and origins of the disease, as well as about decontamination of personal items, such as stethoscopes, and EMS equipment.

In addition, Labrague and colleagues [4] published a systematic review of preparedness levels for future disaster response, but their population of interest was nurses and not EMS personnel. To date, based on our preliminary searches and to our knowledge, there has been no attempt to bring together all the studies that discuss EMS preparedness levels and understand how the evidence translates into practice. Two scoping reviews have been published recently, but one review [5] focused solely on the value of call-centre dispatch and ambulance-based syndromic surveillance for infectious disease detection, whereas the second study [6] only explored applications of quality improvement at public health agencies (not EMS) during the COVID-19 pandemic. Finally, a narrative review [7] investigated the global increase of EMS calls due to COVID-19 and the reasons behind the bottleneck of EMS calls during the early phase of the pandemic. A scoping review of all available evidence on current and past EMS

pandemic preparedness interventions, as well as exploring EMS personnel's experiences and perceptions, would be crucial to identify gaps in the design and implementation of current pandemic preparedness interventions. Furthermore, lessons learnt so far can help provide recommendations for future EMS pandemic preparedness planning.

## Aim and research questions

The aim of this scoping review was to identify and present the available quantitative and qualitative evidence of EMS pandemic preparedness, and how this translates into practice. This included studies of EMS pandemic preparedness plans, intervention implementation and evaluations, and importantly perceptions of EMS staff and patients. The findings of this scoping review will be used to inform future research to strengthen EMS pandemic preparedness planning.

Our research questions were:

1. What interventions (e.g., infection control, PPE) have been implemented within the Emergency Medical Services (EMS) in response to/during pandemics and what outcomes are reported relating to such interventions?

2. What evidence is there describing the experiences of EMS staff and patients during pandemics?

## Methods

We followed the Preferred Reporting Items for Systematic reviews and Meta-Analyses extension for Scoping Reviews (PRISMA-ScR) Checklist [8] and the Arksey & O'Malley framework [9] for conducting this scoping review. The review protocol was registered with the Open Science Framework (osf.io/2pcy7). See Supporting Information for the completed PRISMA-ScR checklist (S1 PRISMA-ScR Checklist) and a copy of the review protocol (S2 Open Science Framework Protocol Registration).

## Inclusion criteria

Studies were eligible for inclusion if they reported on original (primary or secondary data analysis), quantitative, qualitative or mixed-methods studies within which ambulance service/EMS personnel and patients were engaged (including papers with mixed samples, provided that the majority of the participants were EMS staff members and/or patients) during epidemic or pandemic disease outbreaks. For studies to be considered eligible they had to meet the following inclusion criteria: Participants: Ambulance service or EMS staff, pre-hospital patients, attended by ambulance service or EMS staff during epidemics or pandemics and their relatives; Concept/ phenomena of interest: any type of intervention implemented in response to epidemics or pandemics within prehospital EMS/ambulance services, as well as the experiences of EMS staff and/ or pre-hospital patients during epidemics or pandemics; Context: All global prehospital EMS/ ambulance services; Types of study: Quantitative approaches including, but not limited to, interventional studies, feasibility studies, observational studies (cohort and case control), quasi-experimental studies, cross sectional studies and surveys, As well as qualitative designs including, but not limited to, phenomenology, grounded theory, ethnography, and a generic qualitative approach. Finally, multi-methods studies that met the qualitative and/or quantitative inclusion criteria specified above. Data from multi-methods studies were extracted into the respective quantitative and/or qualitative arm of this review and synthesised accordingly.

Studies were excluded if: they were conducted in the emergency department or hospital; the participants were non-EMS personnel (e.g., hospital nurses, General Practitioners, etc.);

the interventions were not implemented during an epidemic or pandemic; and/or the papers were published in a language other than English, due to lack of resources for translation of such papers.

## Information sources and search strategy

Electronic database searches were performed in MEDLINE, PubMed, CINAHL, Cochrane Library, PsycINFO (including content from PsycARTICLES), and Web of Science Core Collection. All database searches were supplemented with internet searches (i.e., Google Scholar), and forward and backward citation tracking from the included studies and review articles. PROSPERO was also searched for protocols of existing (completed or ongoing) systematic reviews. Databases were searched from inception to July 2022. The search strategy used in all the above databases was a combination of the following keywords and related terms: ambulance; emergency medical services; and pandemic. The search terms were entered using Boolean operators and truncation. Medical Subject Headings (MeSH) were also employed in forming the search strategy. For the full search strategy used for each of the databases, see Table 1.

## Study selection and data charting

All references were reviewed and screened independently by seven reviewers (working in pairs, with one reviewer [DL] forming part of more than one pair of reviewers). Titles and

**Table 1. Search strategy.**

| Database | Search strategy | Temporal coverage of each database |
|---|---|---|
| MEDLINE, CINAHL, PsycINFO (via EBSCOhost) | (Emergency Medical Technicians [MeSH] OR | 1868 –present, |
| | Ambulance* OR Emergency Medical Service* OR Pre-hospital OR Paramedic* OR "Out of hospital" OR ems OR "first responder*") AND (Disease outbreaks [MeSH] OR Outbreak* OR Pandemic* OR Epidemic* OR "infectious disease outbreak*" OR "disease outbreak*" OR influenza) | 1943 –present, 1933 –present |
| Web of Science Core Collection | ('Emergency Medical Technicians' OR Ambulance* OR 'Emergency Medical Service*' OR Pre-hospital OR Paramedic* OR 'Out of hospital' OR ems OR 'first responder*') AND (Pandemic* OR Epidemic* OR 'infectious disease outbreak*' OR Outbreak* OR influenza) | 1900 –present |
| Cochrane Library | (Emergency Medical Technicians [MeSH] OR Allied health personnel [MeSH] OR Ambulance* OR Emergency Medical Service* OR Pre-hospital OR Paramedic* OR 'Out of hospital' OR ems OR 'first responder*') AND (pandemics [MeSH] OR epidemics [MeSH] OR Disease outbreaks [MeSH] OR Outbreak* OR Pandemic* OR Epidemic* OR 'infectious disease outbreak*' OR 'disease outbreak*' OR influenza) | 1996 –present |
| PubMed | ("Emergency responders" [MeSH] OR Ambulance* OR Emergency Medical Service* OR Pre-hospital OR Paramedic* OR "Out of hospital" OR ems OR "first responder*") AND ("infectious disease transmission, patient to professional/prevention and control" [MeSH Major Topic] OR "Influenza, human" [MeSH Major Topic] OR Outbreak* OR Pandemic* OR Epidemic* OR "infectious disease outbreak*" OR "disease outbreak*" OR influenza) | 1951—present |

Note: Databases were searched up to July 2022

abstracts were initially screened for relevance and final eligibility was assessed through full-text screening against the inclusion criteria, using a pre-designed study selection form. Any disagreement between the reviewers over the eligibility of references was resolved through discussion between the entire team of reviewers.

A standardised, pre-piloted form was used to extract data from the included studies for data synthesis. Extracted information included: study details (title, authors, date, country), methods (aims, objectives, research questions, study design, setting, data collection methods, intervention characteristics,), participant characteristics (demographics, inclusion/exclusion criteria, method of recruitment, sample selection, sample size), and study findings (main and secondary outcomes, data analysis, conclusions). One reviewer extracted data and a second reviewer checked the data extractions for accuracy. Any discrepancies were resolved through discussion.

### Data synthesis

A narrative synthesis of all quantitative eligible studies was performed and structured around the study design/methodology adopted and aims, key findings, as well as intervention type and content, where appropriate.

Qualitative data from the qualitative studies were also synthesised narratively and presented thematically, according to meaning and content.

## Results

The search strategy identified a total of 22,599 citations and of these 90 were included in this scoping review. Fig 1 presents a flowchart illustrating the results of the selection process.

Out of the 90 included studies (summarised in Tables 2–9), 71 were purely quantitative and 12 were purely qualitative studies. In addition, four studies were mixed-methods [10–13] that involved both a quantitative and qualitative design. One of these studies [11] used interviews only to describe the model for the deployment process of EMS procedures and, as such, only the quantitative arm of this study will be included in this review. In addition, only the qualitative arm of the study by Petrie and colleagues [12] will be included in this review, as their quantitative arm did not meet our inclusion criteria. Similarly, only the qualitative arm of the study by Vilendrer and colleagues [13] will be included in this review, as this study did not report on any relevant demographic and user data, due to restrictions in their data use agreements. Findings of the study by Alwidyan [10], the fourth mixed methods study, will be presented and discussed under the quantitative and qualitative sections of the results below, depending on which type of design is being presented.

Two studies [14, 15] conducted surveys with open-ended questions and described their design as a cross-sectional [15] or mixed-methods [14] study, but analysed their findings qualitatively. One study [16] was a reflection/text and opinion paper. The study characteristics and results of these three studies [14–16] will be presented and discussed as qualitative studies.

Seventy four out of the 90 included studies (82.2%) were conducted and published since the COVID-19 pandemic had started.

### Quantitative synthesis

**Study characteristics.** The 73 quantitative studies (including the quantitative arms of two mixed-methods studies) (Tables 2–8) were published between 2004 and 2022 and were from the USA (n = 24; 32.9%) or Canada (n = 2; 2.7%), Europe (n = 21; 28.8%), Asia (n = 22; 30.1%), and Australia (n = 3; 4.1%), while one Delphi study [17] included an international panel of experts (1.4%).

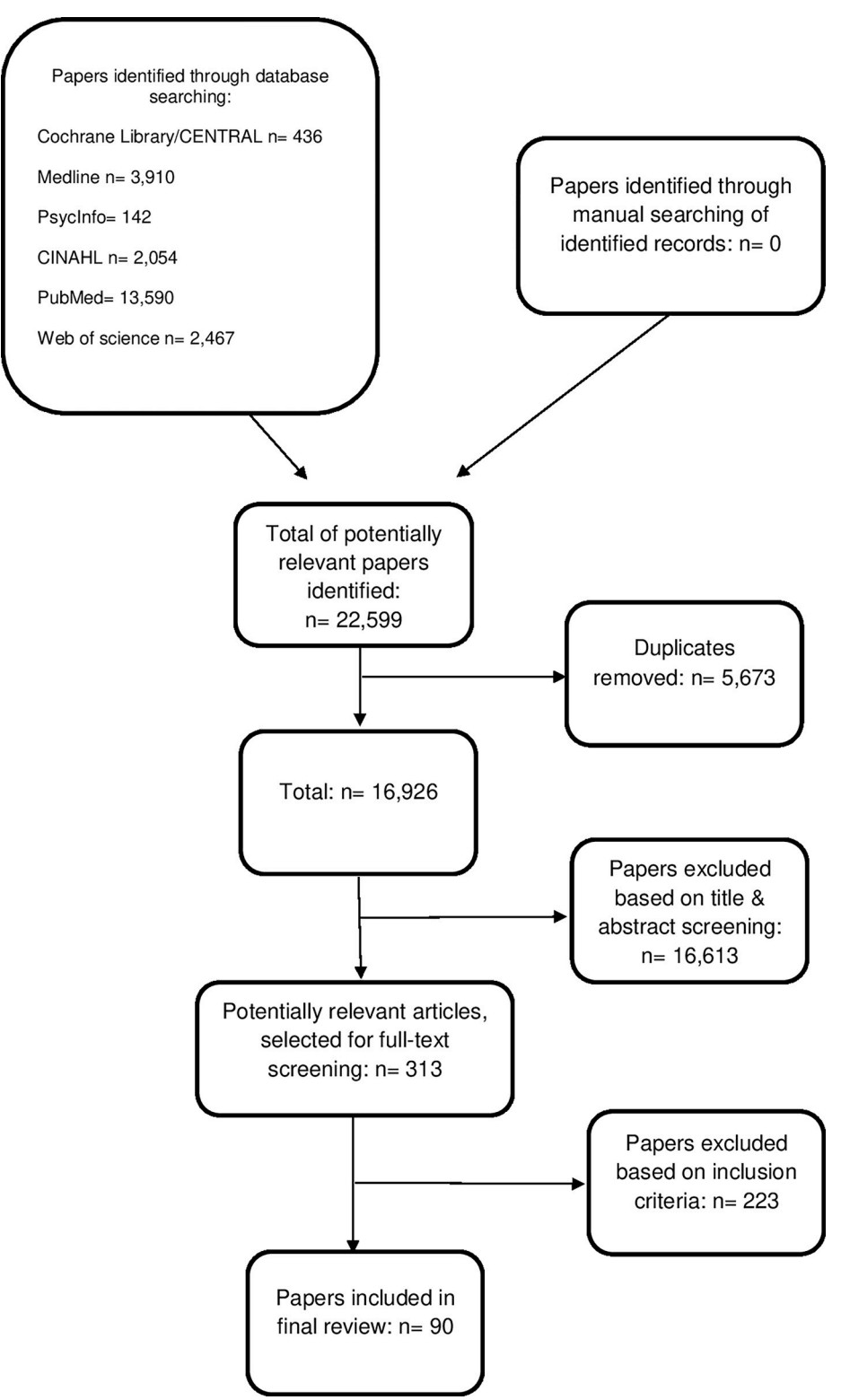

**Fig 1. Flow diagram of study selection.**

**Table 2. Study characteristics of quantitative studies–Willingness to work, treat patients and get vaccinated.**

| Study | Study aims | Study design | Sample | Intervention | Findings |
|-------|-----------|--------------|--------|--------------|----------|
| Alwidyan, 2018; USA | To assess EMS providers' views about working during disease outbreaks and to explore the factors that may influence their willingness to continue working during such situations | Cross-sectional, questionnaire study (Doctoral dissertation) | 40 paramedics & 54 EMTs; 72 male (76.6%), 22 female (23.4%) | N/A | Most participants were willing to report for work, despite having various reasons for concern (e.g., becoming infected and getting ill or dying, infecting family members, shortage in PPE supplies, no known effective treatment or vaccine for infected patients, etc.) |
| Alwidyan et al., 2020; Jordan | To assess the perception and attitude of EMS providers toward working during disease outbreaks, and the factors that may influence their decisions to work or not | Cross-sectional, questionnaire study | 43 EMTs, 86 EMT-intermediate, 330 Paramedics; mean age of 27 years; 325 male, 138 female | N/A | Most participants were willing to come to work during disease outbreaks. The main predictors of reporting for duty were confidence that employers will provide adequate PPE, having adequate knowledge and training for disease outbreaks, agreeing with the strict disciplinary actions to enforce reporting for duty, feeling obligated to work even if some co-workers become infected, feeling obligated to work if they did not receive appropriate training, concern for their family, or being concerned about shortage in PPE |
| Barnett et al., 2010; USA | To identify factors that affect EMS workers' response willingness in the face of a pandemic threat | Cross-sectional, questionnaire study | 586 EMS workers; 34% paramedics & 66% EMTs; 63.2% were above 35 years; 34.1% female (vs male) | N/A | Most participants were willing to report to work, if required or if asked, but not required. Rates of willingness to report for duty dropped if there was a possibility for disease transmission to family members. Being willing to work was related to knowing their responsibilities in a pandemic influenza emergency, being prepared to perform their responsibilities, perceiving that one's response role is important, self-perceived knowledge about the public health impacts of pandemic influenza, and confidence about safety at work |
| Ganter et al., 2021; Germany | To establish a concept for a smartphone alerting system during the COVID-19 pandemic and to evaluate whether it can safely be operated in pandemic conditions | Observational | 571 first responders; 68% male | The restart of a smartphone alerting system in case of emergencies with suspected cardiac arrest, together with providing every first responder with appropriate personal protective equipment (PPE) (N95 mask, protective gown, safety glasses, gloves, bag and mask with an airway filter, a mouth and nose protection to cover the patients face) | Willingness to respond to alarms was lower during the pandemic without PPE and remained lower than before the pandemic even when the volunteers had been equipped with PPE. Most participants were willing to perform chest compressions, defibrillate using an automated external defibrillator, ventilate a patient using a bag and mask and an appropriate airway filter, and ventilate a patient using a face mask |

(*Continued*)

**Table 2.** (*Continued*)

| Study | Study aims | Study design | Sample | Intervention | Findings |
|---|---|---|---|---|---|
| Mackler et al., 2007; USA | To determine if fear of infection would compromise first responders' ability to care for persons potentially infected with smallpox | Cross-sectional, questionnaire study | 95 paramedics; average age was 32 years; 56 were male (59%) | N/A | Most participants would not remain on duty if there were no vaccines and no PPE available. Even if PPE was available but the vaccine was unavailable, only 39% of participants would remain on duty. 91% of paramedics would remain on duty if they were fully protected, but this number dropped to 38% if the respondent believed that their immediate family was not protected |
| Nohl et al., 2021; Germany | The aim of this study was to evaluate the COVID-19 vaccination acceptance of EMS personnel as front-line health care workers in Germany | Cross-sectional, questionnaire study | 1,296 front-line care workers; 283 female, 1,013 male | N/A | 57% of participants were willing to be vaccinated & 27.6% were undecided. Those who were male, had a higher medical education level, were older and felt burdened by the pandemic, were more likely to get vaccinated |
| Rebmann et al., 2020; USA | To Identify determinants of EMS personnel's willingness to work during an influenza pandemic | Cross-sectional, questionnaire study | 433 staff members, 278 EMTs (64.2%) & 155 Paramedics (35.8%); 386 male, 45 female; 318 White (76.8%), 69 Black (16.7%), 6 Asian (1.4%), 6 Hispanic (1.4%); 15 Other/ mixed (3.6%) | N/A | More EMS staff members were willing to work when required versus when only requested (88.2% vs 76.9%). Predictors of willingness to work were believing it is their responsibility to work, believing their co-workers were likely to work, receiving prophylaxis for themselves and their family members, & feeling safe working during a pandemic |
| Roberts & Bryce, 2015; Canada | To examine the overall pandemic preparedness levels of paramedics with emphasis on knowledge about infection prevention and control, actual practices and compliance with existing guidelines, policies and procedures and intent and willingness to report for duty during a pandemic | Cross-sectional, questionnaire study | 370 staff members: 42 Emergency Medical Responders (11%), 288 Primary Care Paramedics (78%), 28 Advanced Care Paramedics (7.6%), 9 Critical Care Paramedics (2.4%), 3 Other (<1%); female 120 (32%), male 250 (68%) | N/A | Most respondents had the intent and willingness to report for work during a pandemic, despite the fact that knowledge was marginal, compliance with practices moderately low, and confidence in EMS pandemic preparedness less than optimal |
| Rutkow et al., 2014; USA | To test if and how USA state emergency preparedness laws (i.e., ability to declare a public health emergency; requirement to create a public health emergency plan; priority access to health resources for responders) can improve EMS staff's willingness to respond during an influenza pandemic | Cross-sectional, questionnaire study | 421 EMS staff members, 258 were EMTs (n = 258, 61%); majority older than 36 years of age (59%); 300 male (71%) | N/A | EMS workers in states that allowed the government to declare a public health emergency (compared to those in states that did not allow the government to declare a public health emergency) were more likely to report that they were willing to respond during an influenza pandemic. However, this difference was not statistically significant |

(*Continued*)

**Table 2.** (Continued)

| Study | Study aims | Study design | Sample | Intervention | Findings |
|---|---|---|---|---|---|
| Tippett et al., 2010; Australia | To investigate the association between knowledge and attitudes of Australian emergency prehospital medical care providers regarding avian influenza and their anticipated behavioural responses during pandemic conditions | Cross-sectional, questionnaire study | 725 emergency prehospital medical care workers | N/A | Almost half of the participants would be unwilling to work during pandemic conditions, one-quarter would not be prepared to work in PPE, & one-third would refuse to work with a colleague exposed to a known case of pandemic human influenza. Willingness to work during a pandemic & willingness to change roles increased with adequate knowledge about infectious agents, whereas refusal to work with exposed or potentially exposed colleagues decreased with adequate knowledge about infectious agents. Confidence in the employer's capacity to respond appropriately to a pandemic increased employee willingness to work & change roles during a pandemic, preparedness to wear PPE. It also decreased the likelihood of refusing to work with colleagues exposed to (suspected) influenza |

Notes: EMS = Emergency medical services; EMT = Emergency medical technician; N/A = Not applicable; PPE = personal protective equipment

A large proportion of studies (n = 15; 20.5%) were cross-sectional, questionnaire studies. Sample sizes (excluding those analysing EMS call volumes, dispatches and/or response times) ranged from 10 to 15,339 participants ([18] did not specify their final sample size). One study [19] only included male participants and many studies had either data missing or did not report gender data. Ages ranged from 19 to over 80 years, with many studies not reporting any relevant data. Only 8 studies (11%) included details on their participants' ethnicity, with the majority identifying as White.

Most studies included combinations of paramedics and Emergency Medical Technicians (EMTs) or other types of EMS personnel (such as nurses or physicians), while some studies defined their participants as emergency prehospital medical care workforce [20, 21], EMS personnel [22, 23] or first responders [24, 25]. One study [17], using a systematic Delphi procedure, included a multidisciplinary group of experts on outbreak preparedness. Various studies included only patients, while two studies included patients and paramedics [26, 27]. Two studies recruited healthy volunteers [28, 29].

**Quantitative narrative synthesis.** After considering each included study's aims, outcomes, and major findings, we developed seven overarching themes, describing EMS pandemic preparedness plans and interventions implemented in response to pandemics.

*Willingness to work, treat patients and get vaccinated.* Seven studies (9.6%) [2, 10, 20, 30–33] asked EMS staff members about their views on working during disease outbreaks/pandemics and factors influencing their decisions. Most respondents would be willing to report for duty in case of a disease outbreak or pandemic, with corresponding percentages ranging from 56.3% [20] to 93% [2]. One further study [24], conducted since the COVID-19 pandemic begun, found that willingness to respond to alarms was lower during the pandemic (with or

**Table 3. Study characteristics of quantitative studies–Preparedness to face pandemics/recommendations.**

| Study | Study aims | Study design | Sample | Intervention | Findings |
|---|---|---|---|---|---|
| Belfroid et al., 2017; Netherlands-based | To capture the views of first responders on what they consider key recommendations for high quality preparedness and identify the recommendations with the highest urgency from the perspective of first responders | Systematic Delphi procedure | A national and two international expert panels | N/A | Article presented 18 recommendations for future pandemic preparedness that related to the development of the preparedness plan and exercises; the evaluation of the level of preparedness; the training of healthcare professionals; sharing information with other organizations; the provision of resources; the installation of an outbreak control group; an access plan for stockpiling and distribution; the plan to expand the healthcare organization's capacity; and the collaboration with regional key stakeholders. |
| Bovim et al., 2021; Norway | To provide decision support for hospital management when preparing for a state of pandemic, by estimating the number of beds that must be present in the ED, the number of ambulances required to maintain response times & the additional number of (boarding) beds required in the ED due to delayed transfer to the COVID-19 ward, due to the lack of available beds in the COVID-19 ward | Simulation study | N/A | Three discrete event simulation models are described to evaluate the resource requirements during the peak of the pandemic | A strict testing policy increased the bed requirements in the ED, while it had the opposite impact on ambulances. Two distinct mechanisms causing boarding time were found, where the effects from boarding time were most prominent during night and weekends |
| Gibson et al., 2020; USA | To investigate available resources, PPE availability, sanitation practices, institutional policies, and opinions among EMS professionals in the USA amid the COVID-19 pandemic | Cross-sectional, questionnaire study | 192 EMS professionals (including EMTs, paramedics & EMS-registered nurses) | N/A | EMS providers had limited access to N95 respirators, received little or no benefits from COVID-19 related work, and reported no institutional policy on social distancing practices. Those who had access to N95 respirators, 31% reported having to use the same mask for 1 week or longer. A third of the participants were unsure of when a COVID-19 patient is infectious. Regular decontamination of EMS equipment after each patient contact did not happen regularly |
| Jadidi et al., 2019; Iran | To evaluate the efficacy and preparedness of EMS in Islamic Republic of Iran to face Ebola | Cross-sectional, questionnaire study | No relevant data reported | N/A | The average score related to preparedness levels was higher than standards. There was no significant difference between the country regions regarding the preparedness of to detect, protect & respond to Ebola outbreaks |
| Roberts & Bryce, 2015; Canada | To examine the overall pandemic preparedness levels of paramedics with emphasis on knowledge about infection prevention and control, actual practices and compliance with existing guidelines, policies and procedures and intent and willingness to report for duty during a pandemic | Cross-sectional, questionnaire study | 370 staff members: 42 Emergency Medical Responders (11%), 288 Primary Care Paramedics (78%), 28 Advanced Care Paramedics (7.6%), 9 Critical Care Paramedics (2.4%), 3 Other (<1%); female 120 (32%), male 250 (68%) | N/A | Knowledge was marginal, compliance with practices was moderately low, and confidence in EMS pandemic preparedness was less than optimal. The majority of respondents, however, intended and were willing to report for work during a pandemic |

*(Continued)*

**Table 3.** (Continued)

| Study | Study aims | Study design | Sample | Intervention | Findings |
|---|---|---|---|---|---|
| Verbeek et al., 2004; USA | To describe the loss of paramedic availability to Toronto EMS during a SARS-1 and SARS-2 outbreak | Prospective observational study | All paramedics of Toronto EMS | A dedicated paramedic surveillance and quarantine program of contact tracing, quarantine, & medical surveillance of paramedics | Work quarantine optimised the number of days on which paramedics were available for duty. Many paramedics developed SARS-like symptoms without being diagnosed as having SARS. This program was able to successfully manage the paramedic resource during a SARS outbreak |

Notes: EMS = Emergency medical services; EMT = Emergency medical technician; N/A = Not applicable; PPE = personal protective equipment

without PPE). Despite that, most participants were willing to perform chest compressions, defibrillate using an automated external defibrillator, ventilate a patient using a bag and mask and an appropriate airway filter, and ventilate a patient using a face mask.

Most papers identified predictors of reporting for duty, such as operating in a state that had emergency preparedness laws [34]; first responders knowing and being prepared to perform their responsibilities in a pandemic [2, 32]; knowing that one of their colleagues had been exposed to suspected or a known case of pandemic human influenza [20]; confidence about safety at work [2, 10, 20, 32] or that the employer would provide appropriate training, an effective treatment and vaccine when available [10] or adequate PPE [30, 31]; receiving prophylaxis for themselves and their family members [31, 32]; having adequate knowledge and training for disease outbreaks [2, 20, 30]; being concerned about self or family safety [10, 20, 30, 33]; having family prepared to function in their absence [10]; lack of confidence in emergency health preparedness and lack of PPE availability [33]; and, believing their co-workers were likely to work [32]. Other predictors of a greater likelihood of reporting for duty were younger age, male gender, single status, and having no young children [31].

Finally, one study [35], exploring the COVID-19 vaccination acceptance of EMS personnel, found that 57% of participants were willing to be vaccinated and 27.6% were undecided. Participants who showed higher willingness to be vaccinated tended to be male, of higher education level, older age, and felt more strongly that they were personally burdened by the pandemic.

*Preparedness to face pandemics/recommendations.* Three cross-sectional, questionnaire studies (4.1%) [3, 18, 33] found that EMS workers exhibited low levels of knowledge and training about infectious diseases, as well as compliance with practices (such as selecting and removing PPE) [3, 33]. Often, they had limited access to PPE equipment and regular decontamination of EMS equipment after each patient contact was not a regular practice [3]. On the contrary, a study by Jadidi and colleagues [18] revealed that Iranian EMS' efficacy and preparedness levels to face Ebola were higher than standards, as represented by factors such as triage, diagnosis, isolation processes, using PPE, as well as transporting and providing care during transfers.

One study [36] used discrete event simulation models to evaluate the resource requirements during the peak of the pandemic, by estimating number of beds needed in the ED, number of ambulances required to maintain pre-pandemic response times for emergency patients, as well as to study the effects of ED boarding time for COVID-19 patients. They found that a strict testing policy increased the bed requirements in the ED, while it led to decreased ambulance response times. They also showed that when boarding is considered, the effects were most prominent during night and weekends.

**Table 4. Study characteristics of quantitative studies–Knowledge & education.**

| Study | Study aims | Study design | Sample | Intervention | Findings |
|---|---|---|---|---|---|
| Albright et al., 2021b; USA | To explore the predictive value of an emergency infectious disease surveillance tool for detecting COVID-19 and the impact of positive screening on using PPE | Retrospective chart review of prehospital care reports and hospital electronic health records | 13,399 patients were transported. 4,329 patients had a positive COVID-19 screen & 263 patients had a positive COVID-19 test | Administering a standardized screening tool, the Emerging Infectious Disease Surveillance Tool from the International Academies of Emergency Dispatch | EMS personnel wore full PPE in 55.7% of patient encounters, did not wear PPE in 8.0% of encounters, & did not document it in 27.9% of encounters |
| Cash et al., 2021; USA | To investigate awareness of and training in PPE during the COVID-19 pandemic and to determine factors associated with reporting these outcomes | Cross-sectional design | 15,339 EMS personnel; the mean age was 39 years; 72% were male; 88% were non-Hispanic/White | N/A | We found high awareness of N95 respirators & air purified respirators (APR) or powered APR (PAPR) use, but less so for N95 fit testing. There was an improvement after CDC guidance for N95 respirator fit testing & APR/PAPR training. Part-time employment, providing 9-1-1 response service, working at a non-fire-based EMS agency, & working in a rural setting were associated with lower odds of awareness/training |
| Davidson et al., 2022; Australia | To develop a simulation software environment to conduct prehospital research during the COVID-19 pandemic on paramedics' teamwork and use of mobile computing devices, and to investigate its feasibility for use as a research and training tool | Pilot implementation and feasibility study | N/A | SPECTRa (Simulated Prehospital Emergency Care for Team Research), an online simulated prehospital environment that lets participants care concurrently for single or multiple patients remotely | The pilot implementation demonstrated that SPECTRa was feasible for use as a research & training tool |
| Eisinger et al., 2009; USA | To determine EMS personnel's knowledge and attitudes regarding infection prevention and control (for potential pandemic influenza) | Cross-sectional, questionnaire study | 141 EMS workers, 56.4% were EMTs and the rest were paramedics; 69% were male (one respondent didn't answer this question) | N/A | PPE was inconsistently used. Glove use was the most consistent protective practice indicated by both EMTs & paramedics, while most neglected to choose gowns for any type of illness. |
| | | | | | Both groups indicated they always perform hand hygiene after patient contact. |
| | | | | | Practices employed to alter the environment within the ambulance were also inconsistent, while there was a consistent lack of disinfection of the ambulance at any time by either group (participants, however, said that their ambulances were routinely cleaned). |
| | | | | | Both groups felt confident in their abilities to protect themselves, but few felt very confident in their abilities |

*(Continued)*

**Table 4.** (Continued)

| Study | Study aims | Study design | Sample | Intervention | Findings |
|---|---|---|---|---|---|
| Gershon et al., 2010; USA | To assess the effectiveness of a multi method pandemic preparedness training intervention | Quasi-experimental study | 129 EMS workers including EMT's and paramedics; mean age was 38.15; 91 male (71.1%), 37 female (28.9%) | A preparedness training intervention (lasting 1 month) that focused on basic influenza knowledge (including routes of transmission and signs/symptoms) and department-specific infection control policies and procedures (especially the proper use of N95 respirators). | There was a significant increase in knowledge & behavioural intentions to use respirators, get vaccinated and report to duty during a pandemic |
| Le et al., 2018; USA | To explore the depth of US EMS practitioners' highly infectious disease training and education | Cross-sectional, questionnaire study | 2,165 EMS staff | N/A | Most participants were aware that their agency had standard operating guidelines for highly infectious diseases & correctly identified routes of exposure for such diseases. Most respondents indicated no maximum shift times in PPE |
| McNally et al., 2021; USA | To determine the impact of virtual simulation to teach EMS personnel respiratory failure management and to explore their perceptions of this learning experience in comparison to other training modalities | Simulation study and survey | 90 EMS personnel underwent the virtual simulation on respiratory failure and 42 of them then participated in the survey | Each session was virtually conducted by a physician. The physician facilitator was remotely broadcasted to the EMS team, performing tasks on a mannequin in the physician's broadcasted room as dictated by the EMS team and providing vital signs. Each session was approximately 25 minutes with 15 minutes of case progression and 10 minutes of debrief | Participants felt more comfortable in managing respiratory failure in both suspected/known COVID-19 patient and non-COVID-19 patients after their sessions. They also found the video platform easy to use, and the most common technical difficulty involved audio issues |
| Rebmann et al., 2020; USA | To Identify determinants of EMS personnel's willingness to work during an influenza pandemic | Cross-sectional, questionnaire study | 433 staff members, 278 EMTs (64.2%) & 155 Paramedics (35.8%); 386 male, 45 female; 318 White (76.8%), 69 Black (16.7%), 6 Asian (1.4%), 6 Hispanic (1.4%); 15 Other/mixed (3.6%) | N/A | A quarter received no pandemic training and only 14.3% had participated in a pandemic exercise. Participants were more willing to work when required & less so when only requested. Predictors of willingness to work when requested were: believing it is their responsibility to work, believing their co-workers were likely to work, receiving prophylaxis for themselves & their families, and feeling safe working during a pandemic |
| Roberts & Bryce, 2015; Canada | To examine the overall pandemic preparedness levels of paramedics with emphasis on knowledge about infection prevention and control, actual practices and compliance with existing guidelines, policies and procedures and intent and willingness to report for duty during a pandemic | Cross-sectional, questionnaire study | 370 staff members: 42 Emergency Medical Responders (11%), 288 Primary Care Paramedics (78%), 28 Advanced Care Paramedics (7.6%), 9 Critical Care Paramedics (2.4%), 3 Other (<1%); female 120 (32%), male 250 (68%) | N/A | Knowledge was marginal, compliance with practices was moderately low, and confidence in EMS pandemic preparedness was less than optimal. The majority of respondents, however, intended and were willing to report for work during a pandemic |

(*Continued*)

**Table 4.** (Continued)

| Study | Study aims | Study design | Sample | Intervention | Findings |
|---|---|---|---|---|---|
| Suppan et al., 2020; Switzerland | To evaluate whether a gamified e-learning module can improve the rate of adequate PPE choice by prehospital personnel in the context of the COVID-19 pandemic | Randomised-controlled trial | 176 EMS personnel; median age was 34 for the intervention & 35 for the control groups; 28 females in the intervention group & 32 in the control group | A previously described gamified e-learning module created under Storyline 3 (Articulate Global) was used in this study. The module contains 19 sections and embeds 7 video sequences. There are two quizzes, a preintervention quiz designed to establish the participants' baseline knowledge regarding PPE, their use and indication, and a postintervention quiz to assess whether these parameters had changed. Questions designed to assess PPE choices are preceded by short clinical scenarios. | Adequate choice of PPE increased significantly after the intervention. Confidence in the ability to use PPE was maintained in the e-learning group, but decreased in the control group |
| Suppan et al., 2020b; Switzerland | To evaluate the impact of a gamified e-learning module on adequacy of PPE in student paramedics | Randomised-controlled trial | 98 EMS student paramedics; median age was 25 for the intervention & 26 for the control groups; 49% females in the intervention group & 39% in the control group | The interactive module was created under Articulate Storyline 3 (Articulate Global, New York, USA). Gamification was used for donning and doffing sequences. Users were asked to drag pieces of PPE in the right order onto the photograph of an EMS provider and to rebuild these sequences for three different COVID-19 risk settings: procedures carrying a high-risk of COVID-19 transmission (i.e., intubation), suspected or confirmed COVID-19 with no need to perform a high-risk procedure, and no clinical suspicion of COVID-19. The whole module was designed to be completed in less than 15 min. Participants had to complete a pre- and post-interventions set of questions assessing their knowledge levels and any changes due to the intervention. | Adequate choice of PPE increased significantly. The e-learning module was of greatest benefit in the subgroup of student paramedics who were actively working in an ambulance company, but not for those not actively working in an ambulance service |
| Tippett et al., 2010; Australia | To investigate the association between knowledge and attitudes of Australian emergency prehospital medical care providers regarding avian influenza and their anticipated behavioural responses during pandemic conditions | Cross-sectional, questionnaire study | 725 emergency prehospital medical care workers | N/A | Willingness to work during a pandemic significantly increased with adequate knowledge about infectious agents, whereas refusal to work with exposed/potentially exposed colleagues significantly decreased |

*(Continued)*

**Table 4.** (Continued)

| Study | Study aims | Study design | Sample | Intervention | Findings |
|---|---|---|---|---|---|
| Visentin et al., 2009; Canada | To assess the knowledge of, use of and barriers to the use of PPE for airway management among EMTs during and since the 2003 Canadian outbreak of Severe Acute Respiratory Syndrome (SARS) | Cross-sectional, questionnaire study | 230 EMS staff (63% were certified at the EMT-paramedic level); most were 30–39 years of age (45.6%); 77.6% male | N/A | Most participants answered correctly 2 or 3 out of 5 questions of the knowledge-based testing |
| Watt et al., 2010; Australia | To investigate knowledge and attitudes of EMS providers in relation to a potential human influenza pandemic and to determine predictors of these attitudes | Cross-sectional, questionnaire study | 421 emergency prehospital medical care workforce; mean age was 43.1 years; 462 male (63.9%), 261 female (36.1%) (2 missing cases) | N/A | Participants had poor knowledge in relation to avian influenza, influenza generally, & infection transmission methods. Less than 5% of the participants felt that they had adequate education/training about avian influenza |

Notes: EMS = Emergency medical services; EMT = Emergency medical technician; N/A = Not applicable; PPE = personal protective equipment

One further study [37] developed and evaluated a dedicated paramedic surveillance and quarantine program, during a Severe acute respiratory syndrome (SARS) outbreak. They determined the number of paramedics on quarantine each day, the type of quarantine, and the development of SARS-like symptoms, and concluded that their program could provide a useful means to managing the paramedic resource during that and any future SARS outbreaks.

Finally, Belfroid and colleagues [17] employed a systematic Delphi procedure and presented 18 recommendations for future pandemic preparedness.

*Knowledge & education.* Thirteen studies (17.8%) [20–22, 32, 33, 38–45] explored issues around EMS staff members' pandemic knowledge and training.

Most studies [21, 32, 33, 40, 45] found low levels of pandemic-related training and knowledge including that of infection transmission. Most studies [33, 40, 45, 46] found that PPE use was inconsistent and knowledge about PPE requirements low, e.g., how N95 masks worked or the correct PPE removal sequence. Practices employed to alter the environment within the ambulance, such as ensuring the desired airflow when transporting a patient with an airborne illness or disinfection of the ambulance at any time were inconsistent, despite respondents reporting that ambulances were routinely cleaned [40].

Only two studies (2.7%) [38, 42] reported adequate levels of training on patient and practitioner safety related to infectious and communicable diseases (including routes of exposure) [42]; how to screen and provide emergency medical treatment for such patients [42]; and respirator use during the COVID-19 pandemic [38]. The main barriers to adequate awareness/training included part-time employment, providing 9-1-1 response service, working at a non-fire-based EMS agency, and working in a rural setting [38]. Some participants did express an interest in having more quality training and not feeling confident enough to respond to diseases with the magnitude or severity of Ebola virus disease [42].

In addition, four studies (5.5%) [22, 41, 43, 44] investigated the effectiveness of various educational interventions. These interventions were effective in increasing knowledge and behavioural intentions to use respirators, get vaccinated and willingness to report for duty during a potential pandemic [41]; increasing adequate choice of PPE [43, 44], especially for those student paramedics who were also actively working in an ambulance company [44]; and increasing comfort levels in managing respiratory failure in suspected/known COVID-19 patients, as well as non-COVID-19 patients [22].

**Table 5. Study characteristics of quantitative studies–Infection risks & control.**

| Study | Study aims | Study design | Sample | Intervention | Findings |
|---|---|---|---|---|---|
| Chen et al., 2021; China | To explore the influence of negative pressure ambulances on the containment of COVID-19 | Observational | N/A | The infection control protocol followed included negative pressure ambulances (equipped with three devices: negative pressure generator, ventilation equipment and isolation stretcher), strict disinfection requirements, as well as a fast supply of hospital beds (e.g., the Huoshenshan hospital, equipped with 1000 beds, was completed in only 10 days) and negative pressure ambulances | Negative pressure ambulances had positive influence on the containment of COVID-19. Newly confirmed cases decreased dramatically |
| Cheng et al., 2021; Taiwan | To conduct a high-fidelity simulation with a specially modified mannequin to determine the most frequently contaminated areas of an ambulance and EMS personnel transferring COVID-19 patients | Quasi-experimental study | Simulation using a specially modified mannequin, supported by two experienced EMTs | A simulation of transportation and basic life support on ambulances carrying COVID-19 patients, using a specially modified mannequin that spewed a fluorescent solution from its mouth to simulate the droplets exhaled by the patient which could be detected using ultraviolet light illumination | The most frequently contaminated areas of an ambulance in the driver's cabin were the left front door's outer handle, driver's handle, gear lever, and mat. |
|  |  |  |  |  | The most frequently contaminated areas in the rear patient's cabin were the rear door, rear door lining, and handle over the roof. The most frequently contaminated areas before the removal of PPE were the lower chest to the belly area, bilateral hands, lower rim of the gown, & shoes. After the removal of PPE, they were the neck, hands, & legs |
| Murphy et al., 2020; USA | To evaluate 9-1-1 EMS encounters for patients with COVID-19 to assess occupational exposure, programmatic strategies to reduce exposure and PPE use | Quasi-experimental study | Among the 274 EMS encounters with patients with COVID-19, there were 429 responding units, involving 700 unique EMS providers with a total of 988 EMS provider encounters | Implemented interventions: enhanced PPE use, classifying long-term care facilities as high-risk locales (requiring full PPE, such as mask, eye protection, gown, and gloves), and implementing a 'scout programme' in which initially one or two EMS providers entered the 'hot zone' to perform the initial in-person evaluation, which then informed the need for remaining EMS crew to don PPE to assist. In addition, when attending cardiac arrest cases and cases requiring aerosol-generating therapies, EMS providers were required to wear eye protection, gown, gloves, and N95 masks | Use of full PPE was 67% & none of those who tested positive were due to occupational exposure from inadequate PPE. Programmatic changes were associated with a reduction in exposures. |
|  |  |  |  |  | Less than 0.5% of EMS providers experienced COVID-19 illness within 14 days of occupational encounter. Programmatic strategies were associated with a reduction in exposures |
| Sadeghi et al., 2021; Iran | To assess the possible factors associated with increasing risk of COVID-19 among EMTs | Case-control study | 66 confirmed cases and 148 healthy EMTs; all were male; mean age of Case group (n = 66) was 35.72, mean age of Control group (n = 148) was 35.09 | investigated various factors associated with increasing risk of COVID-19 among EMTs, such as possible exposure to COVID-19 patients (through work or personal life), PPE use, precautions taken during the patient transfer, knowledge about disinfection processes and post-exposure measures, as well as general precautions during their daily life (e.g., washing hands or social distancing, etc.). | Factors correlated with a higher risk of catching COVID-19 were: having two EMTs taking care of patients, working with a confirmed case teammate, considering the precautions such as seal check after wearing the mask, covering the hair with a medical hat, not using personal items despite protective clothing & avoiding contact with clothing while removing PPE |

*(Continued)*

**Table 5.** (Continued)

| Study | Study aims | Study design | Sample | Intervention | Findings |
|---|---|---|---|---|---|
| Wilson et al., 2021; USA | To estimate infection risks for first responders in ambulances caring for COVID-19 and non-COVID-19 patients through the aerosol and fomite routes and explore the benefit of cloth masks worn by patients, respiratory protection worn by first responders, and masks or respirators worn | Simulation study | N/A | The researchers developed two models/scenarios: initially transporting a COVID-19 patient (Scenario 1), and subsequently transporting an uninfected patient in the same ambulance where there are still aerosols containing COVID-19 and contamination on surfaces from deposition of aerosols from the previous transport (Scenario 2). They also tested different sub-scenarios using various respirator and cloth mask use patterns for both the crew and the patients | Predicted mean infection risks were reduced when simulated first responders wore respirators, the patient wore a cloth mask, and when first responders and the patient wore respirators or a cloth mask. Simulations showed that face masks worn by patients can reduce surface transmission by reducing viral deposition on surfaces |

Notes: EMS = Emergency medical services; EMT = Emergency medical technician; N/A = Not applicable; PPE = personal protective equipment

**Table 6. Study characteristics of quantitative studies–Making improvements regarding resources & PPE.**

| Study | Study aims | Study design | Sample | Intervention | Findings |
|---|---|---|---|---|---|
| Hunt et al., 2022; USA | To determine the efficacy and safety of a novel negative pressure helmet in mitigating simulated infectious particle spread in varied EMS transport platforms during aerosol-generating procedures | Open-label study of the efficacy of the AerosolVE helmet and filtration system | 15 healthy volunteers; 12 males & 3 females | A novel negative pressure helmet. The device (AerosolVE Helmet; Inspire Rx LLC; Ann Arbor, Michigan USA) consists of a helmet equipped with an air purifying filter, based on the common powered air purifying respirator (PAPR), but is re-engineered to reverse air flow | The helmet showed efficacy in creating a negative pressure environment & provided significant filtration of simulated respiratory droplets |
| Hunt et al., 2022b; USA | To test the efficacy and safety of a novel negative pressure procedural tent in mitigating simulated infectious particle spread in varied EMS transport platforms during aerosol-generating procedures | Open-label study of the efficacy of the AerosolVE BioDome system | 15 healthy volunteers; 12 males & 3 females | The tent (AerosolVE BioDome; Inspire Rx LLC; Ann Arbor, Michigan USA) is a clear plastic drape on a collapsible frame that can be secured to a back board or stretcher. It is fitted with a high-efficiency particulate air (HEPA) filter connected to a fan motor, allowing ambient air to be pulled into the tent and passed through the HEPA filter before being released back into the ambient environment (which creates a negative pressure environment within the tent) | The tent showed efficacy in creating a negative pressure environment and workspace around the patient and provided significant filtration of simulated respiratory droplets |
| Kienbacher et al., 2022; Austria | To evaluate the influence of PPE with different types of filtering face piece (FFP) masks on attention and dexterity of EMS personnel during basic life support | Prospective triple-cross over randomized controlled non-inferiority trial | 48 paramedics; mean age 28 years old; 4 females (8%) | Teams of paramedics completed three 12-min long basic life support scenarios on a manikin after having climbed three flights of stairs with equipment, each in three experimental conditions, without pandemic PPE, with PPE including a FFP2 mask with an expiration valve, and with PPE including an FFP2 mask without an expiration valve | Attention and dexterity were significantly better after each scenario, with no differences noted between groups |

*(Continued)*

**Table 6.** (Continued)

| Study | Study aims | Study design | Sample | Intervention | Findings |
|---|---|---|---|---|---|
| Kramer et al., 2022; Germany | To evaluate a compact plasma system regarding its disinfection efficiency inside an ambulance car | Lab-based experimental study | N/A | A portable plasma system developed by the company Plasmatreat GmbH (Steinhagen, Germany) was used for all trials. The developed plasma device is based on a dielectric barrier discharge and operates with ambient air as process gas | Reductions of spores were found on all surfaces and positions within 2 hours |
| Małysz et al., 2020; Poland | To compare 3 chest compression methods used by paramedics wearing PPE during resuscitation in the COVID-19 pandemic | Simulation study | 67 paramedics; median age was 30 (27–33) years; 25 women (37.3%) | The study compared 3 chest compression methods regularly used by paramedics: manual chest compression, chest compression with the TrueCPR feedback device, and chest compression with the LUCAS 3 mechanical chest compression device | Chest compressions with LUCAS 3, compared with manual chest compressions, as well as the TrueCPR, essentially increased the quality of the chest compressions (the depth of the compressions, the rate and chest recoil were more frequently correct when using LUCAS 3) |
| Tsukahara et al., 2020; Japan | To evaluate the feasibility of a portable transparent vinyl chloride shield for use in an ambulance | Feasibility study | 10 EMTs | This study tested the shield (which can be reusable after disinfection with hypochlorite and ethanol) during tracheal intubation with video laryngoscopy, insertion of a laryngeal tube, and manual ventilation using a bag-valve mask (BVM) | The intubation success rate was 100% for all trials, but the average intubation time under the shield was slightly longer than without the Shield. EMTs found intubation with the shield to be feasible and patients felt that the shield did not interfere with BVM ventilation in the ambulance or endotracheal intubation in the ED |

Notes: ED = emergency department; EMS = Emergency medical services; EMT = Emergency medical technician; N/A = not applicable; PPE = personal protective equipment

Finally, one study [39] showed that a simulation software environment (SPECTRa) provided a feasible alternative approach to live prehospital simulation and showed potential for remote healthcare research and training during the COVID-19 pandemic.

*Infection risks & control*. Three studies (4.1%) focused on the safety of ambulances and their role in EMS crew COVID-19 infections [47–49]. Their results showed that negative pressure ambulances can help decrease the number of new, confirmed COVID-19 cases [47]; that the most frequently contaminated areas in ambulances were the left front door's outer handle, driver's handle, gear lever, and mat, the rear door, rear door lining, and handle over the roof [48]; as far as personnel is concerned, the most frequently contaminated areas before the removal of PPE are the lower chest to the belly area, bilateral hands, lower rim of the gown, and shoes, and after PPE removal, traces of fluorescence were observed over the neck, hands, and legs [49]; and that when both crew and patient wore respirators or a cloth mask during a simulated ambulance transfer, these practices reduced predicted mean infection risks by 85% (which was a higher reduction than when only one of them wore a respirator or a cloth mask) [49].

Two further studies (2.7%) [19, 27] examined various factors associated with COVID-19 infection risks in EMS staff members. Results showed that EMS providers' positive tests for COVID-19 (after having been exposed to patients with COVID-19) were not attributed to occupational exposure from inadequate PPE and that programmatic strategies were associated with a temporal increase in adequate PPE use and a decrease in EMS provider exposures [27]. Results also showed that the factors mostly correlated with the increasing risk of COVID-19 in EMTs were having two EMTs taking care of patients, working with a confirmed case

teammate, using personal items (e.g., mobile phone or jewellery) despite protective clothing, contact with the outer surface of clothing while removing PPE, not taking precautions such as seal check after wearing the mask, and not covering the hair with a medical hat [19].

*Making improvements regarding resources & PPE.* Six studies (8.2%) evaluated the use of improved resources or PPE during COVID-19 [28, 29, 50–53]. Three of these studies [28, 29, 53] found that negative pressure devices (a powered air purifying respirator helmet called AerosolVE; a procedural tent called AerosolVE BioDome; and a portable, reusable, transparent vinyl chloride shield together with suction to generate negative pressure, respectively) can

**Table 7. Study characteristics of quantitative studies–Call volumes, ambulance response times & triage.**

| Study | Study aims | Study design | Sample | Intervention | Findings |
|---|---|---|---|---|---|
| Bell et al., 2021; UK | To study patient and staff acceptability and the safety of the decision-making process of video triage | Service evaluation using routine and bespoke data collection (surveys to patients & activity logs kept by clinicians) | 1073 triage calls & 40 patients | Video consultations for 999 calls managed by the clinical hub within the emergency operations centre. Video consultations, using either AcuRX™ or Good-SAM™ | A successful video triage call was achieved in 59.7% of all cases. Video triage improved clinical assessment and decision making compared to telephone alone. Patients were also satisfied with the intervention & had a lower rate of re-contacting the service within 24 hours (compared to telephone triage alone). |
| Bendimerad & Drias, 2022; Saudi Arabia | To study the distribution of ambulances during the rise of Covid-19 cases | Mathematical (computer) modelling | N/A | Novel Deep Self-Learning Approach applied to Artificial Orca Algorithm and based on mutation operators | The Deep Self-Learning approach managed well the dispatching of emergency vehicles during the COVID-19 pandemic |
| Bendimerad et al., 2022b; Saudi Arabia | To explore artificial intelligence approaches to organize the response to emergency calls while keeping the studied area sufficiently covered to possible future calls | Mathematical (computer) modelling | N/A | Two Swarm Intelligence approaches, Artificial Orca Algorithm (AOA) and Elephant Herding Optimization (EHO), were applied to organize the response to emergency calls | Swarm intelligence approaches and especially EHO are able to manage the dispatching of emergency vehicles while respecting the cover of calls during COVID-19 |
| Ganter et al., 2021; Germany | To establish a concept for a smartphone alerting system during the COVID-19 pandemic and to evaluate whether it can safely be operated in pandemic conditions | Observational | 571 first responders; 68% male | The restart of a smartphone alerting system in case of emergencies with suspected cardiac arrest, together with providing every first responder with appropriate personal protective equipment (PPE) (N95 mask, protective gown, safety glasses, gloves, bag and mask with an airway filter, a mouth and nose protection to cover the patients face) | Willingness to respond to alarms was lower during the pandemic without PPE & remained lower than before the pandemic even with the use of PPE. The alarm response rate remained at approximately 50% during the second wave of the pandemic |
| Glober et al., 2022; USA | To evaluate safety of a new EMS protocol directing non-transport of low-acuity patients during the COVID-19 pandemic | Retrospective analysis of data | 222 patients; mean age 40.8 years old; 53.2% male; 55.4% African American | Non-transport protocol in two stages, guided by prevalence of COVID-19 in the community, hospital resource utilization, & EMS run volume. There were distinct paediatric and adult protocols for each stage, designed to identify patients thought to be at low risk for acute decompensation based on co-morbidities, vital signs, and signs and symptoms elicited from patient history. A dedicated EMS medical oversight radio channel was established to allow crews to speak with an EMS physician, the agency's EMS medical director or a base hospital familiar with the protocol | There were large deviations from the novel non-transport protocol. Several patients were admitted to the hospital within 72 hours of non-transport both when the protocol was used correctly, and when it was used incorrectly |

*(Continued)*

**Table 7.** (Continued)

| Study | Study aims | Study design | Sample | Intervention | Findings |
|---|---|---|---|---|---|
| Jaffe et al., 2020; Israel | To describe the role of EMS in a pandemic outbreak, including the dedicated MDA COVID-19 tele-triage centre and home testing by paramedics | Retrospective analysis of data | EMS call volumes and dispatches | The centre was staffed 24 hours per day, seven days per week by EMS dispatchers, public health representatives and volunteers, as well as information technology (IT) specialists who provided IT support. All call takers and dispatchers were given training in initial triage for signs suspicious for COVID-19. A flowchart was programmed into the command-and-control system, together with information on endemic areas and infected patient routes, to help call handlers to identify suspicious cases. If exposure was confirmed and there was a clinical picture suspicious of COVID-19, the details were transferred to the physician for a final decision whether to send a paramedic to collect and send the samples to a laboratory for testing. If COVID-19 was then confirmed, an ambulance team was sent to transport the patient on a dedicated negative-pressure hooded bed to the dedicated COVID-19 sections of the emergency departments. The public could also access the "My MDA" application, which was updated with a self-use interactive questionnaire based on the flowchart, so that in many cases one did not need to call the emergency number | Implementing the changes helped slow the spread of disease by enabling suspected patients to avoid hospitals/community clinics & by managing the treatment of mild cases of COVID-19 patients at home, allowing hospitals to be used for severe cases only |
| Jaffe et al., 2022; Israel | To describe the role of Israel's national emergency prehospital medical organization, in the pre-exposure period, before widespread governmental action | Descriptive study | 121 protected "suspected COVID-19" transports | Dispatcher identifying medical emergency telephone calls, from either individuals or medical sources, as "suspected COVID-19", based on symptoms and travel exposure. Data were also collected for travellers approaching the border checkpoint at the Airport | The total number of protected transports during this time was 121. Of these, 36.3% were referred by medical sources & 63.7% were identified as "suspected COVID-19" by dispatchers.<br><br>The border checkpoint was accessed by 156 travellers: 87 were sent to home-quarantine; 12 were transported to the hospital; 18 were refused entry; and 39 required no further action |
| Jensen et al., 2021; Denmark | To present call volume and two measures implemented to handle the increased call volume to the Copenhagen EMS during the COVID-19 pandemic | Observational | EMS call volumes and queue times | A coronavirus EMS support track that could be reached by those calling the main 1813 medical helpline, as a separate queue system, and was staffed by people with a healthcare-related educational background. The staff followed a protocol resulting in the algorithmic placement of responses and callers were given advice on actions or self-isolation or were directed to the emergency EMS track for triage (a web-based self-triage system) | Total call volume (EMS, 1-1-2 emergency line & medical helpline) increased. The 1813 medical helpline queue time also increased. There was no correlation between call volume and web triage usage |
| Karunathilake et al., 2022; Sri Lanka | To evaluate an integrated patient management system where patients could access a medical professional through a short messages service (SMS) & calling system | Short report of an evaluation | Over 70,000 patients & over 30,000 SMSs | A virtual triaging system for identification and evacuation of those who needed hospitalization and to facilitate home management for mild and asymptomatic patients. This service provided telephone triage, patient advice, and coordinated with the national ambulance system to evacuate ill patients | The numbers of oxygen-dependent patients and deaths declined rapidly, & the number of available beds increased. Implementing this intervention helped to bring the crisis under control |

(*Continued*)

**Table 7.** (Continued)

| Study | Study aims | Study design | Sample | Intervention | Findings |
|---|---|---|---|---|---|
| Kim et al., 2021; South Korea | To study the impact of COVID-19 on EMS processing times & transfers to ED among patients with acute stroke symptoms before and during the COVID-19 pandemic | Retrospective observational study | 1570 pre COVID-19 patients (mean age 68 years old; 52.3% male) & 1441 COVID-19 patients (mean age 69.4 years old; 47.7% male) | In the prehospital stage, EMS providers were recommended to use PPE on all dispatches. In the hospital stage, the ED strengthened the screening of patients with COVID-19-related symptoms and pre-emptive isolation through triage & prohibited the allocation of additional beds in the ED | The total number of patients using EMS for acute stroke symptoms decreased, while EMS processing time increased. There was a significant decrease in the number of patients transferred to an ED with a comprehensive stroke centre & an increase in the number of patients transferred to Eds nearby |
| Kyriacou et al., 2021; Cyprus | To present the design of Emergency Response protocols and the development and implementation of a secure emergency handling platform that was created to support the Ambulance Department of the Ministry of Health of Cyprus | Observational | 112,414 cases during the previous 25 months, out of which 4,254 were Covid-19 cases | A secure emergency handling platform to aid triage, ambulance dispatch and incident handling from paramedics | This system was successfully used for 25 months and was a useful tool in dealing with Covid-19 |
| Lee et al., 2022; South Korea | To study the usability, feasibility, acceptability, and appropriateness of the information and communication technology for emergency medical services (ICT-EMS) systems to improve the transportation of emergency patients during COVID-19 | Prospective, questionnaire study | 187 EMTs; median age 31; 55.6% female | Novel ICT-EMS systems to triage patients at the scene & decide the appropriate destination hospital for critical patients, and for patients with suspected COVID-19 | Participants had an overall negative perception of the ICT-EMS systems. 26.7% of participants agreed that ICT-EMS implementation was possible, appealing, and suitable |
| Marincowitz et al., 2021; UK | To assess how accurately NHS 111 telephone services identified those who suffered an adverse outcome needing an emergency response (suspected COVID-19); and to identify any factors that may have affected the accuracy of telephone triage in identifying suspected COVID-19 | Observational study | 40,261 adults who contacted 111 telephone triage services; 62% were White, 10.8% were Asian/Asian-British, 2.1% Black/Black-British, 1.3% Mixed, 1.5% other ethnic groups | From 18 February 2020 onwards, NHS England advised patients with suspected infection to contact the National Health Service (NHS) 111 service (instead of attending healthcare providers), which is a UK, free-to-use 24-hour telephone triage service for urgent health problems | Telephone triage had 74.2% sensitivity & 61.5% specificity for identifying suspected COVID-19. Respiratory comorbidities may be overappreciated, & diabetes underappreciated as predictors of deterioration. Repeat contact with triage service was also an important under-recognised predictor of deterioration, with 2 or more contacts associated with false negative triage |
| Marrazzo et al., 2020; Italy | To quantify how the first public announcement of confirmed COVID-19 in Italy affected a metropolitan region's EMS call volume and how rapid introduction of alternative procedures at the public safety answering point managed system resources | Cross-sectional design | EMS call volumes and dispatches | The main implemented modifications (involving primarily staffing and COVID-19 tailored algorithms) were augmenting staffing (both call handlers and ambulance crew), referring healthy callers to public health information call centres; using algorithms for detection, isolation, or hospitalisation of suspected COVID-19 patients; and, sending specialised medical teams to the public safety answering point for triage and case management, including ambulance dispatches or alternative dispositions | There was a huge increase in calls, but ambulance response intervals and dispatches were successfully contained |

*(Continued)*

**Table 7.** (Continued)

| Study | Study aims | Study design | Sample | Intervention | Findings |
|---|---|---|---|---|---|
| Mulyono et al., 2021; Indonesia | To evaluate the emergency response operations during the COVID-19 pandemic | Agent-based modelling approach with input parameters from interviews with EMS staff | Response times & 2 interviews with the head of the Emergency Services and a paramedic | Special procedures were put in place, including the crews wearing PPE & disinfecting it and the ambulance before and after use | Four factors contribute to response times. The process of preparing crew/ambulance during the pandemic & coverage area significantly contributed to the response time. Traffic density & crew responsiveness were less significant |
| Ng et al., 2021; Singapore | To study the potential impact of the widespread restrictions and changes related to COVID-19 on EMS' utilisation as well as attendant out-of-hospital cardiac arrest and prehospital return of spontaneous circulation rates | Retrospective analysis of data | EMS call volumes and response times | The 'circuit-breaker' measures that were taken in response to the pandemic were: having all ambulance crew wearing appropriate PPE (i.e., N95 mask, goggles, gown, hair and shoe covers, and gloves) when attending a suspected COVID-19 patient; stopping the activation of fire bikes and fire appliances to cardiac arrest cases to minimise exposure and cross-contamination between frontline emergency response crews; stopping the activation of community first responders through Singapore Civil Defence Force' myResponder mobile application; and continue having operations centre specialists manning the national 995 hotline to advise callers to perform dispatcher assisted cardiopulmonary resuscitation (hands only). | EMS call volume & total out-of-hospital cardiac arrests remained comparable to pre-pandemic years |
| Park et al., 2022; South Korea | To investigate whether emergency medical centres for critical care (EMC-CC) had been effective in reducing ambulance diversions & shortening prehospital times for patients | Retrospective observational study | 219,763 EMS responses; 50.8% male | The triage system was reinforced and the capacity of general beds and isolation rooms in Eds were monitored in real time to identify available hospitals. The EMC-CC gave priority to seriously ill patients with suspected COVID-19 within the region, while patients with mild symptoms were referred to other emergency medical institutions | Ambulance diversion rated increased and so did all prehospital times, except for the scene time of cardiac arrest patients |
| Perlini et al., 2020; Italy | To investigate the response of the Pavia EMS network to the COVID-19 pandemic outbreak in Italy | Retrospective analysis of data | EMS call volumes and dispatches | An integration between the triage algorithms of the "out-of-hospital" and "hospital" system components. The activation of this new system led to 22 revisions of the triage algorithm in only 5 weeks. They also separated the access to the emergency department and diverted suspected COVID-19 patients to an infective disease outpatient clinic; adapted the hospital triage and diagnostic protocols to optimise patient care; and shared the indications for the diagnostic nasopharyngeal swab and for the immune response assessment with the Virology Unit | There was an increase in the calls to the AREU 112 system. To facilitate a huge increase in number of admissions, the hospital reshaped many general ward Units, which became Covid-19 Units (up to 270 beds) and increased the intensive care unit beds from 32 to 60 |
| Prezant et al., 2020; USA | To understand the impact of the COVID-19 pandemic on NYC 9-1-1 EMS system | Longitudinal study | EMS call volumes and response times | A series of pre-planned strategies were implemented, including a computer-assisted triage system, increasing capacity by using additional local and out-of-state ambulances, and addressing low-acuity call-types by telemedicine referrals or by a treat/release/no-transport option after ambulance response | Call volume (including high-acuity, life-threatening call-types) increased, especially for respiratory & cardiovascular call-types. Response times also increased. Post-surge, fewer calls were received, compared with pre-pandemic times |

(*Continued*)

**Table 7.** (Continued)

| Study | Study aims | Study design | Sample | Intervention | Findings |
|---|---|---|---|---|---|
| Rautenstrauss et al., 2022; Germany | To investigate the benefits of designating ambulances to serve only infected patients and suspected COVID-19 cases | First-stage optimization model/simulation | N/A | A two-stage approach to design and evaluate the ambulance split, including models to pre-select and evaluate (using the approximate Hypercube Queuing Model) ambulance splits. | An ambulance split did not reduce the average response time for the examined data set (call volume & the infection probability were too low). However, a sensitivity analysis showed that long isolation times and high infection probabilities can make an ambulance split beneficial for patients and EMS personnel, as an ambulance split reduces the average response time without significantly increasing the mean infection probability for EMS personnel |
| Riyapan et al., 2022; Thailand | To explore the impact of COVID-19 on the prehospital management and outcomes of out of hospital cardiac arrests (OHCA) | Retrospective observational study | 691 patients: 341 (49.3%) in the pre-COVID-19 period & 350 (50.7%) in the COVID-19 period; mean age was 63.07; 60.5% male | There was a change in prehospital OHCA management protocols: the EMS and ED providers were required to don appropriate PPE. Two participating sites allowed their providers to perform endotracheal intubations at the scene & one site recommended that EMS providers avoid advanced airway insertion during the outbreak | There was a significant decrease in the survival rate to discharge, prehospital intubation and prehospital drug administration during the COVID-19 outbreak. The first-responder response interval was significantly longer during the COVID-19 outbreak |
| Saberian et al., 2020b; Iran | To determine the effects of COVID-19 on the workload of EMS Tehran and the associated changes to patient presentation on EMS arrival | Retrospective analysis of data | EMS call volumes and dispatches | Initially an Advanced Surveillance System of Coronavirus Committee was created, while EMS Tehran extended its operations, increased the number of EMS staff (and limited the amount of time off they could take), provided them with more PPE and essential supplies, trained them in screening for and diagnosing COVID-19, and added coronavirus-consulting phone lines to answer patient questions | EMS calls & dispatches increased. The average time on telephone hold & length of call decreased. There were significant increases in chief complaints of fever & respiratory symptoms |
| Sechi et al., 2020; Italy | To assess if Business Intelligence (BI), which allows access to real-time information, can enable front line managers to promptly make informed decisions to drive system improvements in critical times such as an epidemic | Data mapping/a business intelligence model study | EMS call volume | The regional EMS Trust used BI daily to track the number of first aid requests received to their emergency lines, as well as to analyse those episodes that were classified (during the telephone dispatch interview) as respiratory and/or infectious, to be able to identify the numerical trend of episodes in each municipality (increasing, stable, decreasing). The system can then decide accordingly how and where to reallocate the resources based on real-time data recorded and elaborated by BI | BI was successfully applied to promptly identify clusters and patterns of the SARS-CoV-2 epidemic |
| Snooks et al., 2021; UK | To describe volume and pattern of calls to emergency ambulance services, proportion of calls where an ambulance was dispatched, proportion conveyed to hospital, and features of triage used during the first wave of the COVID-19 pandemic in the UK | Cross-sectional design | EMS call volumes and dispatches | All ambulance services reported implementing various changes in response to the pandemic in coding calls and protocols for assessment or care, such as upgrading their response to some calls, for example, for reported "ineffective breathing", offering a higher response priority to ST-segment elevation myocardial infarction (STEMI) patients in some clinical groups, as they noted a significant increase in such patients, or introducing a new COVID-19 related question for all calls as a surveillance tool ("Have you or the patient had a high temperature or new continuous cough in the last 14 days?") | Call volume varied widely between services, but all ended the study period with a lower call volume than at baseline. Case mix and workload changed significantly & triage models and prehospital outcomes varied between services |

(*Continued*)

**Table 7.** (*Continued*)

| Study | Study aims | Study design | Sample | Intervention | Findings |
|---|---|---|---|---|---|
| Zahedi et al., 2021; Iran | To test two innovative approaches to design a relief supply chain network to address multiple suspected cases during the pandemic | Internet of Things and mathematical modelling study | EMS response times | Their prioritizing approach aims to minimize ambulance response times, while the allocating approach aims to minimize the total critical response time | Implementing the proposed Internet of Things-based methodology, the results showed 35.54% decrease in the number of confirmed cases |

Notes: BI = Business Intelligence; EMS = Emergency medical services; EMT = Emergency medical technician; N/A = Not applicable; PPE = personal protective equipment

**Table 8. Study characteristics of quantitative studies–Identifying patients, testing & vaccinations.**

| Study | Study aims | Study design | Sample | Intervention | Findings |
|---|---|---|---|---|---|
| Albright et al., 2021b; USA | To explore the predictive value of an emergency infectious disease surveillance tool for detecting COVID-19 and the impact of positive screening on using PPE | Retrospective chart review of prehospital care reports and hospital electronic health records | 13,399 patients were transported. 4,329 patients had a positive COVID-19 screen & 263 patients had a positive COVID-19 test | Administering a standardized screening tool, the Emerging Infectious Disease Surveillance Tool from the International Academies of Emergency Dispatch | The screen had a sensitivity of 74.9% & specificity of 67.7%. The positive predictive value was 4.5% & the negative predictive value was 99.3%. Skilled nursing facility patients were the most common high-risk population identified (19.5%). EMS personnel wore full PPE in 55.7% of patient encounters, did not wear PPE in 8.0% of encounters, & PPE use was not documented in 27.9% of encounters |
| Constantine et al., 2021; USA | To discuss the implementation and deployment of mobile integrated health and community paramedicine testing sites to provide screening, testing, and community outreach during the first months of the COVID-19 pandemic | Retrospective data analysis | Data from 6 test sites and a total of 4,342 patients; 65.6% patients self-identified as White, 21.2% as African American, 1.2% as Asian & 0.7% as Native American; of all patients, 14.3% self-identified as Hispanic & 85.7% as non-Hispanic | Mobile test centre, staffed with EMS personnel. Staff maintained persistent PPE with gown, gloves, and mask with face shield. Gloves were exchanged between every patient contact following CDC guidelines. Additionally, the staff maintained hot and cold zones to separate charting, registration, and equipment maintenance activities from patient testing and evaluation | The 6 test sites evaluated 4342 patients, of which 401 patients (9.2%) had positive test results (62.8% were women). The estimated duration of each encounter was 3 to 5 minutes, where paramedics took a brief history, and did specific physical examination & screening for signs of hypoxemic respiratory failure. There were no cases of accidental exposure or failure of PPE for these paramedics |
| Dillon et al., 2020; UK | To evaluate the predictive value of pre-hospital recorded oxygen saturation levels for death in COVID-19 patients, in an attempt to assist early recognition of COVID-19 and prediction of severe disease | Retrospective service evaluation | 143 patients; mean age was 73 years; 61 male (56%); 31 Black, Asian and minority ethnic 28% | N/A | The lowest recorded pre-hospital oxygen saturation was an independent predictor of death (when controlling for age, gender, and history of Chronic Obstructive Pulmonary Disease), with a 1% reduction in oxygen saturation increasing the odds of death by 13%. In addition, lower pre-hospital oxygen saturation levels predicted death only after adjusting for the pre-hospital NEWS2, as pre-hospital NEWS2 was higher in those who died than in those who survived to discharge. |

(*Continued*)

**Table 8.** (Continued)

| Study | Study aims | Study design | Sample | Intervention | Findings |
|---|---|---|---|---|---|
| Fitzpatrick et al., 2022; UK | To describe and summarise the characteristics of people identified by ambulance service telephone triage as having COVID-19 symptoms | Retrospective record linkage cohort study | 214,082 emergency calls; median age for positive patients 66 years old (50–78), for negative 66 years old (47–80); 49.8% females & 49% females for positive & negative patients, respectively | Three key components were introduced to support pre-hospital care of COVID-19 in Scotland: a modified protocol (Protocol 36) to include specific COVID-19 symptom-focused questions; Covid 'hubs' were introduced in each major Health Board area to enable professional-professional discussions (ensuring the most appropriate pathway for possible COVID-19 patients); a clinical acuity guideline that provided support for clinicians to identify COVID-19 symptoms and high risk patients who would benefit from further professional-professional discussion with a senior physician based in the Covid hub | The positive predictive value of the Protocol 36 to identify potentially COVID-19 positive patients was low (17%). In those identified by Protocol 36 as Covid-19 negative, 30-day mortality was higher in those not conveyed, but in the Protocol 36 Covid-19 positive calls, mortality was higher in those conveyed. Thirty-day mortality rates of those with COVID-19 diagnosed through virology was between 28.8 and 30.2% |
| Kjerulff et al., 2022; Denmark | To study if decompensation during a one-minute sit-stand test is useful for identification of seemingly stable patients suspected of COVID-19, who would later develop a need for oxygen treatment | Quality assurance project with prospectively collected data | 156 patients triaged to ambulance response by general practitioners, of which 86 (55%) were tested with the sit-stand test; median age was 52 years old (35–64); 42 (48.8%) were male | During COVID-19, in stable patients over 70 years of age with no risk factors for serious disease, the one-minute sit-stand test was implemented as precautionary security measure (by ambulance personnel) before considering treating and releasing the patient on-scene. The test consisted of asking the patient to sit and stand as many times as possible during one minute without supporting with the arms. The test was considered to indicate decompensation (i.e., positive test) if the oxygen saturation decreased more than 3 percentage points and/or if the patient could not perform more than 12 sit-stand repetitions in one minute. The test was also considered positive if saturation fell below 90% or if the patient was unable to continue the test. The decision on whether to admit the patient to hospital or not was made by a physician telemedically | 30/86 (34.8%) were older than 70 years and/or had comorbidities for serious disease and were, thus, off-guideline tested. 10 (6%) patients had a positive COVID-19-test. 17 (20%) patients decompensated during the test and of these, 9 (53%) were treated with oxygen. The authors concluded that the sit-stand test may be useful for decision making in emergency settings |
| Levy et al., 2021; USA | To explore whether identified Persons Under Investigation (PUI) for COVID-19 (by EMS) are associated with hospitalizations for COVID-19 disease | Retrospective data analysis | 225,355 emergency calls during the 141-day study period, of which 26,855 patients (12%) were identified as PUIs; most were above 80 years old, female & White, or Black/African American | EMS identifying PUIs for COVID-19 by using the following clinical criteria: shortness of breath, cough, sore throat, muscle aches, new loss of sense of smell or taste, or diarrhoea, regardless of travel history | There was a strong correlation between EMS PUIs and COVID-19 hospitalizations, especially at a 9-day lag from time of EMS PUI transports to new COVID-19 hospitalizations |

*(Continued)*

**Table 8.** (Continued)

| Study | Study aims | Study design | Sample | Intervention | Findings |
|---|---|---|---|---|---|
| Marincowitz et al., 2021; UK | To assess how accurately NHS 111 telephone services identified those who suffered an adverse outcome needing an emergency response (suspected COVID-19); and to identify any factors that may have affected the accuracy of telephone triage in identifying suspected COVID-19 | Observational study | 40,261 adults who contacted 111 telephone triage services; 62% were White, 10.8% were Asian/Asian-British, 2.1% Black/Black-British, 1.3% Mixed, 1.5% other ethnic groups | From 18 February 2020 onwards, NHS England advised patients with suspected infection to contact the National Health Service (NHS) 111 service (instead of attending healthcare providers), which is a UK, free-to-use 24-hour telephone triage service for urgent health problems | Telephone triage had 74.2% sensitivity & 61.5% specificity for identifying suspected COVID-19. Respiratory comorbidities may be overappreciated, & diabetes underappreciated as predictors of deterioration. Repeat contact with triage service was also an important under-recognised predictor of deterioration, with 2 or more contacts associated with false negative triage |
| McCann-Pineo et al., 2021; USA | To estimate the value of an Emergency Medical Dispatch telephone screening process for the identification of hospital diagnosed COVID-19 positive patients | Diagnostic accuracy study | 3,443 encounters during the study time frame, with 652 (18.9%) of patients testing positive for COVID-19; 29.5% were White, 13.3% were Black; of all patients, 9.2% identified as Hispanic and 90.8% as non-Hispanic | The Emerging Infectious Disease Surveillance Tool for Novel Coronavirus, as developed by the International Academies of Emergency Dispatch, was adapted and implemented to identify possible COVID-19 positive patients | Sensitivity of the Emergency Medical Dispatch telephonic screening was estimated as 75.0% and specificity was 45.5%, while the positive predictive value was 24.3% and negative predictive value 88.6% |
| Pozner et al., 2022; USA | To determine how EMS clinicians and agencies have been utilized in vaccination efforts during the COVID-19 pandemic | Retrospective, questionnaire study | Survey responders were EMS medical directors and EMS system administrators. 88 counties in North Carolina, USA, participated in the vaccination efforts (including paramedics & EMTs) | EMS clinicians and agencies were utilized in vaccination efforts during the COVID-19 pandemic | Vaccination efforts included activities such as planning and logistics, non-medical roles, vaccine preparation, medical screening pre-vaccination, vaccine administration, medical observation post-vaccination, & home vaccinations. Reasons cited for not being involved in vaccination efforts (or for limited involvement) included county health departments not needing assistance, vaccine hesitancy amongst clinicians and the politicization of COVID, inadequate staffing, the presence of "robust vaccination clinics" in the community, & staffing shortages. 27 counties used EMTs (37%), 36 used Advanced EMTs (49%), & 73 used Paramedics (99%) |
| Saberian et al., 2020; Iran | To evaluate the efficiency of pre-hospital triage tools in determining the prognosis of probable COVID-19 patients | Diagnostic test accuracy study | Data of 557 probable COVID-19 patients; mean age was 56.93 years; 67.5% were males | The accuracy of 3 prognostic prediction models (qSOFA, NEWS, and PRESEP) designed for pre-hospital systems was evaluated in this study | The sensitivity and specificity for qSOFA were 25% and 85.7%, respectively; for NEWS the sensitivity and specificity were 83.6% and 32.7%, respectively; & for PRESEP the sensitivity and specificity were 54.1% and 55.6%, respectively. The authors concluded that the available prehospital triage tools do not have proper efficacy to predict death, ICU admission, and disease severity of COVID-19 patients |

(*Continued*)

**Table 8.** (Continued)

| Study | Study aims | Study design | Sample | Intervention | Findings |
|---|---|---|---|---|---|
| Saberian et al., 2021; Iran | To evaluate the value of COVID-19 antibody rapid test assessment in EMS personnel | Cross-sectional design | 243 EMS personnel; mean age of 36.14 (range 21 to 59) years; 213 male (87.7%), 30 female (12.3%) | Comparison of the results of chest computed tomography (CT) scan and antibody rapid test in EMS personnel with confirmed COVID-19 and those who were symptomatic and/or asymptomatic but had exposure to a colleague with probable or confirmed COVID-19 | Rapid antibody test could help diagnose COVID-19 in both asymptomatic and symptomatic EMS personnel, but sensitivity could be enhanced when used together with other diagnostic methods, such as RT-PCR test or chest CT-scan |
| Spangler et al., 2021; Sweden | To describe the presentation of Covid-19 patients in a Swedish prehospital care system, and asses the predictive value of Covid-19 suspicion as documented by dispatch and ambulance nurses | Retrospective analysis of data | 11,775 hospital visits with linked prehospital data; median age was 70; 52.8% female | N/A | Out of 11,894 prehospital records, 481 had a primary hospital diagnosis code related to or positive test results for Covid-19. Covid-19-positive patients had worse outcomes than patients with negative test results, with 30-day mortality rates of 24% vs 11%, but lower levels of emergent transport rates of 14% vs 22%. Regarding the assessment of predictive value, sensitivity was 76% and 82% for dispatch and ambulance suspicion respectively, while specificities were 86% and 78% |
| Wooten et al., 2021; USA | To investigate the feasibility of utilizing EMS staff for mass vaccinations | Feasibility study | 16 EMS staff (paramedics & EMTs) | EMS personnel prepared the vaccines for administration and administered them without required oversight | 319 vaccines were administered over a period of 6 clinic days, with 272 given by EMS providers (85%). 58% of the total vaccines were given by EMTs or EMT students. EMTs made up 36% of the workforce, Paramedics made up 16% of the workforce (combined percentage of overall vaccination staffing was 53%) |
| Zmora et al., 2021; Israel | To investigate the efficiency of COVID-19 drive-through test centres | Economic evaluation | Data from 198 stationary and mobile drive-through complexes | 4 stationary drive-through testing complexes in the largest cities in Israel: Jerusalem, Tel Aviv, Haifa, and Beer Sheva, and 8 mobile drive-through testing complexes in remote or other high-volume areas. Vehicles advanced to the sampling position staffed by 2 workers: one with PPE who performed the sampling, and a second one who verified identification labels on the specimens. The sampler put the specimen into a biohazard bag, changed gloves and washed hands between each case | The EMS led drive-through test centres were cost-effective and efficient in performing large numbers of viral tests, when compared with home testing |

Notes: BI = Business Intelligence; EMS = Emergency medical services; EMT = Emergency medical technician; N/A = Not applicable; PPE = personal protective equipment

**Table 9. Study characteristics of qualitative studies.**

| Study | Study aims | Sample | Method of data collection | Method of data analysis | Findings |
|---|---|---|---|---|---|
| Alexander et al., 2020; USA | To examine the attitudes and experiences of emergency medical technicians (EMTs) and paramedics when caring for patients with suspected or confirmed Ebola Virus Disease (EVD); to understand how EMTs and paramedics make the decision whether or not to care for Persons Under Investigation; to make recommendations in preparation for the response to future infectious disease outbreaks | 22 (6 EMTs and 16 Paramedics); 25–52 years (Mean 38yrs); Male 19 (86%), Female 3 (14%); White 16 (73%), Black 1 (4.5%), Hispanic 1 (4.5%), Other 4 (18%) | Three 30 to 60-minute focus groups | A grounded theory approach was used, and data were analysed using inductive content analysis | First responders were more comfortable caring for EVD patients if they had received adequate education & transparency from their administration, which led to a decreased perceived danger of the disease and decreased hesitancy when caring for EVD patients. |
| | | | | | Those that expressed the most hesitancy also expressed the most emotional distress. Recommendations included continuing education, tiered training & peer training models, better communication & collaboration between EMS systems and departments, and the development of an infectious disease response team |
| Alwidyan, 2018; USA (Doctoral thesis) | To assess EMS providers' views about working during disease outbreaks and to explore the factors that may influence their willingness to continue working during such situations | 14 interviews with EMTs and 8 interviews with paramedics, with a total of 22 interviews; Male 15, female 7 | Face-to-face, in-depth semi-structured interviews (lasting 25–72 minutes, average 1 hour) | Inductive approach | Participants were a little concerned about working during disease outbreaks, but this did not prevent them from fulfilling their work obligations (especially when PPE was provided). Decision to come to work during disease outbreaks was influenced by family safety, disease pathogenicity, workplace culture, training, resources, and confidence in their employer. Participants reported having high confidence in their employer to provide training, resources, and other measures to keep the providers and their families safe |
| Belfroid et al., 2018; Netherlands | To understand how healthcare organizations can prepare to meet the needs of their healthcare workers by capturing the experiences of those who dealt with patients with suspected Ebola virus disease | 3 ambulance nurses | Semi-structured, in-depth interviews (lasting 25 min to 1 hr) | Thematic analysis | Most participants felt positive about the experience of caring for a patient with suspected Ebola and felt motivated for further training and simulation exercises for similar outbreaks. Clear protocols were considered very important and pre-arrival briefing helped them to prepare mentally for providing care for a person with suspected Ebola |
| Boechler et al., 2021; North America/ Canada | To explore the lived experiences of the COVID-19 outbreak from a paramedic viewpoint | 424 paramedics | Online survey consisting predominantly of open-ended questions | Descriptive cross-tabulations & inductive thematic analysis | The main theme was the role of leadership and its impact on paramedics' experiences throughout the first wave of the COVID-19 pandemic. Trust and reciprocal communication with management played a large role in determining a practitioner's perception of risk |

*(Continued)*

**Table 9.** (Continued)

| Study | Study aims | Sample | Method of data collection | Method of data analysis | Findings |
|---|---|---|---|---|---|
| Friedman & Strayer, 2020; USA | To describe the New York City EMS Response to COVID-19 | 7 EMS staff members | Reflection/text and opinion paper | None. This paper only presented excerpts of participants | Participants had to make difficult choices, such as choosing between their own safety & the well-being of the patient. Not allowing families to accompany the patient in the ambulance and being unsure whether staff could have done anything more (often due to new protocols they had to implement) weighted heavily on them. Participants also expressed concerns regarding their own families & endangering them. They also reported feeling physically/ mentally drained & ethically challenged. They also felt that there was a lack of organization and communication by leadership, management, and politicians. Others also mentioned the positives, such as being given the authority to act like the clinicians that they were trained to be |
| McAlearney et al., 2022; USA | To learn how first responders feel about COVID-19 and its impact on their work | 21 first responders who were employed at the Divisions of Police (PD) and Fire (FD) (including paramedics); mean age 46 years old (range 29–56); 14 males & 7 females | Semi-structured interviews | Deductive-dominant thematic analysis | Participants were concerned about COVID-19, but where unsure about having a COVID-19 vaccination. They felt increased stress due to the pandemic and especially fear of COVID exposure during emergency responses, infecting family members, and frustration surrounding new work policies. Many participants also felt there was a lot of misinformation on COVID-19 |
| Mohammadi et al., 2021; Iran | To identify strategies to manage COVID-19-related challenges faced by EMS personnel in the south of Iran | 27 pre-hospital emergency care personnel; mean age was 34.3 years | Semi-structured interviews | Content analysis method suggested by Granheim and Lundman (2004) | Participants felt that there is an urgent need for comprehensive care protocols with clear instructions on how to deal with COVID-19 patients, as well as public education on requesting and handling pre-hospital emergency care during a pandemic. Participants also felt that their health depended on access to enough, adequate PPE (for themselves & patients) & allocation of separate ambulances to COVID-19 patients. Participants also felt there is need for reduction/ better management of professional challenges, such as the psychological security of personnel, occupational burnout & stress |

(*Continued*)

**Table 9.** (Continued)

| Study | Study aims | Sample | Method of data collection | Method of data analysis | Findings |
|-------|-----------|--------|---------------------------|-------------------------|----------|
| Oliphant et al., 2022; Canada | To identify what makes paramedics feel safe or at risk while serving on the front lines during COVID-19 and to develop recommendations to support them in their role | 21 paramedics; 11 females and 10 males | Semi-structured interviews | Thematic analysis | Participants described several factors that made them feel exposed to risk during working on the front lines of the COVID-19 public health response, including stress connected to PPE and equipment access, risks of infection to self and family, lack of communications and feelings of being over-exposed and under-considered |
| Olsson et al, 2022; Sweden | To investigate ambulance staff's self-reported hand hygiene perceptions and compliance & to explore whether the COVID-19 pandemic has affected compliance | 204 EMS staff (EMTs, RNs & specialist RNs); 51% were male & 48% female; mean age was 42.3 years | Cross sectional survey, with open-ended questions/free-text answers | Thematic analysis based on the methods proposed by Braun & Clark | Almost all (92% of) participants felt that their hygiene routine compliance had improved during the COVID-19 pandemic. These improvements were mainly due to fear & needing to learn how to deal with the pandemic |
| Parvaresh-Masoud et al., 2021; Iran | To explore EMTs' experiences of the challenges of prehospital care delivery during the COVID-19 pandemic | 15 EMTs; 25–51 years old; all male | Semi-structured interviews | Graneheim & Lundman's five-step conventional content analysis approach | Participants expressed a need for accurate information about COVID-19. Behaviours varied mostly due to conflicting policies and guidelines, which led to people feeling there is reduced risk of COVID-19, which then resulted in non-adherence to guidelines, increased prevalence of COVID-19 and increased EMS contacts. |
| | | | | | Other issues identified were obsessive & inappropriate use of disinfectants, difficult care delivery conditions, fear over the transmission of COVID-19 to self and others, burnout due to heavy workload, ineffective communication with hospital staff, ethical conflicts, lack of PPE & treatment plans, staff shortages, and inadequate support by authorities |
| Petrie et al., 2022; Australia | To explore the mental health symptoms and working environment among Australian paramedics during the COVID-19 pandemic and explore their experiences of work and wellbeing during that time | 95 paramedics; 55 were aged 31–50 years old (57.9%); 48 females (50.5%) | Cross-sectional survey with open-ended free-text qualitative items | A qualitative descriptive approach using content analysis | Main issues reported were about safety and risk in the workplace, uncertainty and upheaval at work (including constantly changing procedures and information) and at home, and lack of crisis preparedness in emergency services |

(*Continued*)

**Table 9.** (Continued)

| Study | Study aims | Sample | Method of data collection | Method of data analysis | Findings |
|---|---|---|---|---|---|
| Rees et al., 2021; UK | To explore paramedic experiences of providing care during the 2020 COVID-19 pandemic and develop theory in order to inform future policy and practice | 18 paramedics; ages ranging 26–60; 11 male, 9 female | One-hour, one-to-one semi-structured interviews | Evolved grounded theory | Most participants expressed concern for themselves and their families during the pandemic and felt a sense of fear and risk while working. Working under such extreme circumstances led to changes to hygiene practices, avoiding visiting friends and relatives, and talking more openly on making preparations for their own death. They often felt conflicted when having to transfer someone to the hospital (feeling it may not be in the patient's best interest). Changes in PPE practices & protocols were frustrating & participants expressed relevant concerns. Participants faced rapid disruption to their personal and working lives & had to adapt accordingly. Participants also discussed how trust in information sources was essential and reported a sense of solidarity across their profession in being united working together to face the pandemic |
| Sarchahi et al., 2022; Iran | To explore the challenges and experiences of EMS in the context of the COVID-19 pandemic in Iran | 15 participants (6 paramedics, 3 dispatchers, 1 anaesthesiology assistant, 2 physicians & 3 nurses); average age of 33.8 years old (range = 24–48 years); 66.7% were male | Semi-structured, in-depth interviews | Content analysis (Graneheim & Lundman approach) | The main Issues discussed by participants were lack of preparedness of EMS for the pandemic; shortage of PPE (especially early in the pandemic) & worries about self & family safety; psychological distress and negative emotions (e.g., fear of getting ill & dying); shortage of staff; & challenges associated with delivering care for patients using PPE (such as profuse sweating, shortness of breath, local pain and damage to ears/face with prolonged use of a mask) |
| Smith, 2018; Australia | To encourage discussion within the paramedic profession and broader community on the role and responsibility of paramedics during disasters | 48 participants, consisting of 19 paramedics and 29 community members; paramedic participants ranged in age from 29 to 52 years, while community participants ranged in age from 35 to 64 years; 32% of paramedics identified as female and 68% identified as male; 68% of community members identified as female and 32% identified as male | Semi-structured interviews and focus groups | Grounded theory | Paramedics and community members agreed that acceptable limitations on paramedic duty to treat during disasters are required, considering factors such as personal health circumstances, pre-existing mental health conditions, competing personal obligations, & unacceptable levels of personal risk |

(*Continued*)

**Table 9.** (Continued)

| Study | Study aims | Sample | Method of data collection | Method of data analysis | Findings |
|---|---|---|---|---|---|
| Smith et al., 2021; Australia | To explore the experiences of people living in Victoria, Australia when utilizing EMS during the COVID-19 pandemic | 67 people living in Victoria, Australia; aged 32–78 years old (average age = 52); 54% were male | Semi-structured interviews | Interviews were analysed using Colaizzi's approach | Participants expressed concerns regarding exposure and infection, which often delayed EMS use, especially among participants with chronic health conditions. Participants with acute health conditions were worried about the impact of COVID-19 on their care but continued to use services when needed. Those who provided care for people with sensory/developmental difficulties identified communicating with EMS workers while wearing PPE as challenging, with face masks being especially problematic for those who were deaf or hard-of-hearing. Finally, children and older people needed clear, repeated communication with EMS staff to ensure that information was appropriately understood (particularly when EMS staff wore PPE) |
| Tawalbeh, 2021; USA (Doctoral thesis) | To evaluate the sources of work-family conflict among EMS providers during the COVID-19 pandemic | 30 EMS providers (paramedics & EMTs) | In depth semi-structured interviews | Analysis followed the three-phase procedure described by Hesse-Biber | There were several challenges and concerns for EMS providers during COVID-19, including strain-based conflicts, time-based conflicts, and behaviour-based conflicts. Participants discussed their fears and concerns, role ambiguity, lack of PPE, inadequate education and training, increased workload & mandatory overtimes, decreased time spent with their family, excessive precautionary measures (i.e., becoming more vigilant and more cautious not to get the virus to their homes), obsessive behaviors in order to avoid catching the virus, negative emotions (e.g., stress, anger), and limited interactions with their family members |
| Vilendrer et al., 2021; USA | To evaluate adoption, acceptability, and appropriateness of a medical app-based intervention (COVID-19 Guide App) designed to support access of first responders and essential workers to COVID-19 information and testing services | 7 EMTs, 4 EMTs with fire training, and 5 physician police/EMS advisors; first responders tended to be male, 30–49 years old, and White | Semi-structured 30-minute interviews using a protocol grounded in the Consolidated Framework for Implementation Research | Thematic analysis | App acceptability was mixed among. Participants emphasised the need for personalized and accurate information, access to testing, and securing personal safety. Challenges related to interprofessional coordination and a "culture of heroism" |

*(Continued)*

**Table 9.** (Continued)

| Study | Study aims | Sample | Method of data collection | Method of data analysis | Findings |
|---|---|---|---|---|---|
| Yilmaz et al., 2022; Cyprus | To investigate the difficulties faced by EMS staff in the transport of patients to hospital during the COVID-19 pandemic, and to determine strategies to cope with these difficulties | 49 EMS workers (15 EMTs, 19 paramedics & 15 doctors); ages ranged from 22 to 59; 40.8% (n = 20) were male & 59.2% (n = 29) were female | Structured interview form | Thematic analysis | The difficulties EMS personnel encountered were social problems (e.g., the unconsciousness of people, social isolation, the overburdening of healthcare workers and the health system by the public, etc.), personal problems (e.g., psychological problems, inability to work comfortably, etc.) & system-based problems (e.g., lack of sufficient supplies, time consuming disinfection processes, excessive shifts, etc.). Participants also identified coping strategies that they used |

Notes: EMS = Emergency medical services; EMT = Emergency medical technician; EVD = Ebola Virus Disease; PPE = personal protective equipment; RN = registered nurse

filtrate or reduce aerosol dispersion and exposure to airborne particles, thus, making ambulances potentially safer for EMS personnel. Another study [51] showed that a compact atmospheric plasma device could be used successfully to disinfect ambulances.

In addition, Małysz and colleagues [52] showed that chest compressions with LUCAS 3 can increase the chest compression quality (when compared with manual chest compressions and the TrueCPR), as evidenced by the depth and rate of the compressions, as well as chest recoil. Finally, one study [50] evaluating the influence of PPE with different types of filtering face piece (FFP) masks on attention and dexterity of EMS personnel during basic life support procedures found that neither of these two neuropsychological components were affected by FFP mask use.

*Call volumes, ambulance response times and triage (logistics of service delivery)*. Twenty five studies (34.2%) [11, 24, 54–76] looked at the impact of the COVID-19 pandemic on EMS utilisation and how implementing a variety of COVID-19-related interventions (such as changing coding calls and protocols for assessment or care; developing coronavirus EMS support tracks or web-based self-triage systems, etc.) affected EMS utilisation.

Data analysis of most studies revealed that despite an initial increased demand on EMS resources (especially in relation to call volume), EMS response remained, on the average, relatively controlled, as demonstrated by overall call volume [67, 75], response times [24, 66], daily ambulance diversion rates [68], and the number of out-of-hospital cardiac arrests [67].

Other studies, however, found less positive results related to a sustained increase of EMS calls [58, 69, 70, 73], queue times (though the significant increase in the total EMS call volume was mitigated by the implementation of a coronavirus EMS support track) [60], EMS dispatches [69, 73], EMS processing times [62], prehospital times (except for the scene time of cardiac arrest patients) [68], and response times/interval [70, 72]. Mulyono and colleagues [11], employing an agent-based simulation model, found that the main factors contributing to increased response times were the process of preparing crew and ambulance during the pandemic (relating to the safety procedure in handling patients), service coverage area, traffic density and crew responsiveness. The recommended coverage area for maintaining a low response time was 5 km.

Twelve studies (16.4%) [54–57, 59, 61, 63–65, 71, 74, 76] explored ways of managing the response to emergency calls (during the COVID-19 pandemic) by implementing a range of technology and protocol/systems interventions. These were: applying Business Intelligence to the management of EMS [74]; using the Internet of Things to design a relief supply chain network to address multiple suspected cases during the pandemic [76]; applying a novel Deep Self-Learning Approach to Artificial Orca Algorithm and based on mutation operators to address ambulance dispatching and emergency calls covering problems [55]; applying two Swarm Intelligence Algorithms (Artificial Orca Algorithm and Elephant Herding Optimization) to organize and manage the dispatching of emergency vehicles while respecting the cover of calls during a crisis [56]; running a first-stage optimization model to designate ambulances to serve only infected patients and suspected cases [71]; and, using information and communication technology for emergency medical services (ICT-EMS) systems to improve the transportation of emergency patients [64]. A few interventions [6.8%] also focused on phone or video triage [54, 59, 61, 63, 65], with positive results. A study [57] evaluating the safety of a new EMS protocol directing non-transport of low-acuity patients during the COVID-19 pandemic reported large deviations from the novel non-transport protocol and that several patients had to be admitted to hospital, both when the protocol was used correctly and when it was used incorrectly.

*Identifying patients, testing & vaccinations.* Ten studies (13.7%) investigated the diagnostic accuracy of EMS in identifying COVID-19 patients [23, 46, 65, 77– 82] or predicting death from pre-hospital vital signs [83].

According to their findings, initial vital signs (with the exception of body temperature) [82], prehospital triage tools [81], and telephone screening processes/surveillance tools used by emergency medical dispatchers [46, 65, 77, 80] had little to moderate predictive value for the identification of COVID-19 patients and/or death.

Other studies (n = 4; 5.5%) have found more encouraging results, such as the lowest recorded pre-hospital oxygen saturation being an independent predictor of mortality in COVID-19 patients [83]; rapid antibody testing helping to diagnose COVID-19 in both asymptomatic and symptomatic EMS personnel [23]; newly developed clinical criteria for identifying COVID-19 patients showing a strong degree of correlation between such emergency transports and new hospitalizations [79]; and, a prehospital sit-stand test identifying stable, suspected COVID-19 patients in risk for later deterioration [78].

Two studies (2.7%) [26, 84] explored how EMS can help facilitate testing for COVID-19 by evaluating the implementation of drive-through COVID-19 testing facilities operated by EMS. Results from both studies showed that COVID-19 testing performed by EMS staff can be efficient and safe to operate for both the staff and the patients [26, 84], as well as cost-effective [84].

Finally, two studies (2.7%) looked at the role of EMS in mass vaccinations and concluded that EMS providers were "uniquely equipped to participate in mass immunization efforts" [85] and played an important role in planning and logistics, patient screening and observation, vaccine preparation and administration, and home vaccination efforts [86].

## Qualitative synthesis

**Study characteristics.** The 18 qualitative studies (including the qualitative arms of 3 mixed-methods studies, the surveys that analysed their findings qualitatively, and the text and opinion paper) (Table 9) were published between 2018 and 2022 and were predominantly from the USA (33.3%). Sample sizes ranged from 3 to 424 participants and included both male and female participants (27.8% did not report any data on gender). Participants were EMS

staff members or community members. One study included EMS providers [16] but did not report any more information about their participants. Two studies [13, 87] also reported the perceptions and experiences of hospital healthcare workers and essential workers (e.g., gas station and grocery store employees), respectively, whose views, however, will not be included in this review, as they did not meet the study's criteria for inclusion.

Only 12 studies (66.7%) presented information about the participants' age, with ages ranging from 22 to 78 years old for emergency services personnel; community members' ages ranged between 35 and 64 years old [88]. Only two studies (11.1%) [13, 89] specified the ethnicity of their participants, with the majority identifying as White.

Most studies were based on individual semi-structured interviews (61.1%) and two studies were PhD theses (11.1%) [10, 90]. One study [16] was a reflection/text and opinion paper that included excerpts of participants but gave no additional information about their data collection methods and did not include any formal analysis. Various methods of analysis were employed, with seven studies using thematic analysis (38.9%).

**Qualitative data synthesis.** We used narrative synthesis to summarise and explain the findings of this scoping review.

*Motivation, confidence, and feelings about working during pandemics.* Working during a pandemic was a traumatic experience for many staff members [16, 91], who felt that they were often faced with significant challenges [13, 16, 91, 92], such as not knowing how to properly treat patients, not feeling safe due to poor PPE or the constant changes in protocols and procedures [16]; being unable to socialise and take comfort in colleagues and friends [12, 16]; as well as having less contact with their families [12, 13, 16, 90].

When asked about their feelings about working during pandemics, EMS personnel reported feeling a myriad of different emotions, such as stress [12, 13, 16, 25, 87, 89–91, 93, 94], anxiety and fear [10, 12, 13, 15, 16, 25, 87–96], as well as feelings of frustration [13, 91, 94] or failure, when they couldn't save a patient [16]. They also felt they were faced with difficult clinical and ethical decisions in their practice, such as having to decide between prioritising their own safety and the safety/care of the patient [92].

Despite these negative emotions, many participants also felt positive about the experience of caring for patients and reported feeling proud [87], excited [87, 89], or even safe and protected [87]. They also mentioned what they felt were the positive outcomes of pandemics, such as giving them a chance to test their protocols and increasing their team spirit [87], as well as encouraging the use of technology across the healthcare system [91] and using what they've learned to help expand prehospital care [16].

As a result, the majority of EMS personnel were willing to report for duty during a pandemic [10, 13, 16, 89] and often felt they had a high professional and ethical obligation to work under any and all situations [10, 13, 16, 88, 89, 91].

There were others who were unwilling to report for work or provide care for patients [10, 88, 89]. Some of the main reasons behind their unwillingness to report for duty were not understanding the cause and modes of transfer of a disease [10], not having been provided with appropriate PPE [10], not feeling confident in their skill set [10], role ambiguity [90], lack of trust in their employer [90], and being concerned about their own or their family's safety [10, 89, 90]. On the contrary, some of the factors that would positively affect someone's willingness to report for duty were receiving adequate education and training, having appropriate PPE and equipment, and transparent protocols [10, 89].

*Balancing safety and risks.* Overall, participants felt that being an EMS staff member was inherently risky work [10, 89, 90, 92, 94] but understood and accepted the risks and felt a duty to perform their jobs despite this [10, 16, 88, 89, 91, 94].

Various participants expressed concerns about their own safety [10, 12, 15, 16, 25, 90–92, 94–96], not having adequate training or PPE [10, 93–95], and the risk of transmitting the disease to their family [10, 12, 13, 15, 16, 25, 87, 89–92, 94–96]. As a result, they often took extra care and precautions to limit their risk of exposure [10, 15, 90]. Similarly, families of EMS personnel were also anxious about their safety and that of their working relative [87]. Finally, one study [97] found that fear of exposure and infection delayed EMS utilisation among patients with chronic health conditions and was also a concern for patients with acute health conditions (although they did continue to access services as required).

Despite participants' need for and greater feeling of safety with PPE, they also discussed the negative side of wearing PPE while providing care for patients (especially during hot weather), such as experiencing stress and anxiety, profuse sweating, shortness of breath, local pain, restricted movement, or discomfort due to fogging goggles and/or prolonged use of masks [90, 95, 96]. Similarly, patients and their carers also discussed how communicating with EMS staff wearing PPE was challenging, especially for those who were deaf or hard-of-hearing [97].

*The limits to personal moral duty.* Even though most participants felt that EMS personnel had a duty to report for work, many also felt that there were potential limitations on duty to treat [88, 91]. Some of these acceptable limitations were their own physical health (such as pregnancy or pre-existing chronic conditions) or that of a family member [88, 91]; having mental health problems [88]; being a single caregiver with dependents or when both parents were healthcare workers [88]; as well as work-related factors, such as lack of appropriate PPE, anti-viral medication or appropriate vaccines during infectious disease outbreaks/pandemics, lack of appropriate quarantine facilities away from the home, and lack of relevant training [88].

One paper [88] also discussed community views on risk-taking by EMS personnel. Community members had differing views; some felt that EMS personnel shouldn't be expected to put themselves at risk to treat patients during a disaster, pandemic, or even on a normal day, whereas others felt that there was a certain level of duty or obligation that comes with being an EMS worker [88].

*Need for information, communication, and support.* First responders expressed a desire for infectious disease information (such as routes of transmission, incubation period, infection rates, policies and protocols) [10, 90, 93, 95], as well as follow-up information regarding a transported patient's health status and detailed, localised data that could help them understand the geographic spread of cases throughout their area [13]. However, many also felt that receiving too much and constantly changing information was overwhelming and challenging [13, 14, 91] and that there was a lot of misinformation and lack of reliable data (in regard to COVID-19), which often made them sceptical about the information they were receiving [13, 16, 25, 90–92].

Participants also highlighted the importance of proper communication during disease outbreaks [10, 12–14, 16, 87, 89–91, 94] and how they felt that it was their employer's responsibility to keep them (and their families) informed with the most up-to-date information about the disease and what they were expected to do [10, 12, 87]. Despite many participants reporting feeling adequately informed by their employers [10, 91], but some felt that there was lack of communication by leadership, management, and politicians [12, 16, 94]. Participants said they would have liked to receive more consistent and transparent information about the disease [12–14, 87, 89–90, 92], the provision of support in case of illness [94], and any protocols for triaging and transporting patients [12, 13, 87, 89, 90, 93, 94]. Others felt that there was lack of communication and collaboration between departments and organisations, such as the emergency department and EMS or the hospital and coordinating centres such as the Centre for Disease Control in the United States [13, 14, 16, 89, 92, 94].

Another topic discussed in various papers was the importance of support from employers, colleagues and managers. Participants felt that building a relationship of trust between colleagues was an important element in the EMS [10, 90] and reported often turning to their colleagues and team managers for emotional support and for stress relief [10, 13, 16, 87]. In contrast, participants found it particularly tough when they could not socialise or take comfort from their teammates, for example during COVID-19 [16].

EMS personnel also recognised that their organisations put effort into their comfort and safety, and knowing that their organisation was continuously reviewing and improving procedures made them feel safe and protected [87]. They also expected organisations to provide them with resources, necessary materials and PPE, training and communication during disease outbreaks [10, 87, 92], as well as prioritising testing to keep them safe and to enable them to return to their duties [13]. In addition, some felt that it was the employer's responsibility to provide a "safe haven" to their families as well, including offering them vaccines and/or treatment, if available [10]. Some EMS workers, however, felt unsupported and "left alone to fend" for themselves [12–14, 16, 94].

Regarding mental health support, opinions were also divided. Some would have liked to have received such support [12, 87, 93], while others felt either that they did not need it or that they could get it from other sources (e.g., family, friends, colleagues, private counsellors, the service Chaplain, etc.) [91]. Finally, EMS personnel found lack of financial support in the case of illness a major challenge, especially for those who were new in their role or couldn't take any sick days, and often led to people hesitating to get tested, as they wouldn't be able to take any time off work [13].

*Requirement for resources, training, guidance, evaluation & solutions.* Participants felt that adequate and up-to-date training and education were important [10, 12, 87, 89, 95] and could influence their willingness to report for work [10, 89]. Most participants were satisfied with the training they received, as they felt it prepared them well for their tasks [87, 89], while others were not satisfied [10, 12, 89, 90, 91], especially when it was lecture-based and not hands-on training [10].

Participants also felt that having clear, transparent, and simple protocols was equally important [87, 89, 90, 93] and that it helped them remain calm by following the instructions [87]. In addition, lack of transparency about how the protocols were designed [89] and the constant protocol changes (e.g., about PPE) as the pandemic evolved, only added to their anxiety, frustration, and confusion [12–14, 16, 90, 91, 93–95] and made them concerned about whether the new protocols were sufficient [87]. Some also thought that these changes in clinical practice and guidance were rushed [91], that the quality of care was being compromised [91] and they felt ethically challenged, as the new protocols were asking them to deviate from what they were taught [16]. Participants also felt that their organisations and the governments were overall ill prepared to face a pandemic, such as COVID-19 [12, 95].

Regarding the provision of resources and equipment to keep them safe and to ensure that they are able to carry out their job properly, various participants reported not having access to appropriate or up-to-date PPE during the COVID-19 pandemic and that this was a major concern for them [12, 13, 90–92, 94–96]. Participants also mentioned that in some cases EMS workers did not comply to wearing PPE, mainly due to force of habit, not being used to having PPE on, not recognizing when it is the appropriate time to use the PPE, and/or thinking they won't need them since they may not directly contact the patient [10]; this study, however, was conducted prior to the COVID-19 pandemic. Concern about and frustration with lack of compliance with hand hygiene routines (during the COVID-19 pandemic) was also discussed [15]. The main factors affecting hygiene compliance were the unpredictable work environment, situations where time is critical, worries about the risk of using new protective equipment (e.g.,

gas masks, which were introduced in the ambulance service during the pandemic), and having initiatives supported by their managers/organisations [15]. Having access to testing was also considered important for workers to be able to return to their duties, but many reported challenges in scheduling testing [13].

Finally, participants made a series of recommendations for future outbreak response. Participants felt that it would be best to have specialised teams dealing with outbreaks, such as a permanent infectious disease response team that would be responsible for maintaining and updating protocols, the training of personnel, and the "institutional readiness of disease outbreaks and epidemics" [89]. Participants also highlighted the importance of continuous training and education for pandemic preparedness [12, 89] and suggested using a tiered training model (or even a peer training approach), where a few selected staff members would be trained comprehensively, who would, in turn, train others based on what their roles would be [89]. They also thought it would be a good idea to have regular "drills for the triage and transport of infectious patients, similar to that of mass casualty and disaster drills", as well as to have training on how to use different types of PPE, to ensure that everyone is always prepared for a future pandemic [89]. Other suggestions were to have a dedicated app, through which accurate, up-to-date information (including localized infection rates and spread) could be delivered directly to EMS personnel, holding local question and answer sessions, or using social media to keep them informed [13].

## Discussion

This scoping review explored the available quantitative and qualitative evidence of EMS pandemic preparedness (i.e., to be able to respond and take action effectively on a personal and organisational level), and how this translates into practice. The findings of the review have shown that the majority of the EMS personnel are prepared and willing to report for duty during pandemics, despite their concerns for their own and their families' safety and the many challenges they are faced with.

More specifically, participants reported being willing to report for work during pandemics and the main factors impacting their willingness were: their levels of training for and knowledge of disease outbreaks, and confidence in their skills; feeling that it's their responsibility to work; confidence about safety at work (including adequate PPE, availability of treatment and vaccines); being concerned about their own or their family's safety; and mistrust with the employer. Results also showed that the participants' knowledge was often marginal (especially about infection transmission mechanisms and use of PPE), they had limited training on pandemic response, and either had issues accessing appropriate PPE or using PPE consistently. Lack of proper disinfection of the ambulances was also reported. Past studies have also shown moderate levels of perceived preparedness for the next pandemic, especially in reference to training and confidence in skills, of both social workers in hospital settings [98] and nurses [4]. In our review, various educational interventions were found to be effective in improving participants' knowledge levels and intentions to use PPE, as well as improving management of respiratory failure in COVID-19 patients and could be considered appropriate means of educating EMS personnel on future pandemic preparedness issues.

Numerous studies also looked at infection risks and control and found that negative pressure ambulances can help reduce numbers of new COVID-19 cases; which areas in an ambulance are the most frequently contaminated areas during transfer of a patient with an infectious respiratory disease (including which areas of the EMS workers' body are most frequently contaminated before and after the removal of PPE); the ability of face masks worn by both crew and patients to reduce predicted mean infection risks; that proper use of appropriate

PPE can decrease occupational exposure; a portable, reusable, transparent vinyl chloride shield for use in an ambulance, together with suction to generate negative pressure, can reduce aerosol dispersion and exposure to airborne particle (without interfering with ventilation in the ambulance or endotracheal intubation in the emergency department); and, that chest compressions with LUCAS 3 can increase the chest compression quality. Also, the factors mostly correlated with the increasing risk of COVID-19 in crew members were having two crew members taking care of patients, working with a confirmed case teammate, using personal items (e.g., mobile phone or jewellery) despite protective clothing, contact with the outer surface of clothing while removing PPE, not taking precautions such as seal check after wearing the mask, and not covering the hair with a medical hat.

In addition, many studies looked at the impact of the COVID-19 pandemic on EMS utilisation and found that, although there was an initial increase in call volumes (and sometimes response times), on average, most ambulance dispatches remained relatively controlled (or even decreased in some cases, especially during later waves of the pandemic). Often this was the result of the different interventions that were implemented after the pandemic had started, such as having a coronavirus support track or using Business Intelligence models to identify infection clusters and relocate vehicles and personnel accordingly to these areas where it was more needed, to name a few. A past scoping review [5] on the utility of emergency call centre, dispatch, and ambulance data for syndromic surveillance of infectious diseases also concluded that data timeliness, high level of data standardization, and the clinical value of call-centre dispatch and ambulance data can help detect infectious disease outbreaks.

Other studies explored the diagnostic accuracy of EMS personnel in identifying patients or predicting death from pre-hospital vital signs, as well as how EMS can help facilitate testing for COVID-19 and mass vaccinations. Accordingly, results showed that initial vital signs (with the exception of body temperature) and prehospital triage tools (qSOFA, NEWS, NEWS2 and PRESEP) have little predictive value for the identification of COVID-19 patients or death, intensive care unit admission, and disease severity of COVID-19 patients, respectively. In addition, it was shown that rapid antibody test can help diagnose COVID-19 in both asymptomatic and symptomatic EMS personnel, but sensitivity could be enhanced when used together with other diagnostic methods, such as RT-PCR test or chest CT-scan. Finally, it was shown that EMS staff are uniquely equipped to perform COVID-19 testing and participate in mass immunization efforts efficiently and safely, for both the staff and the patients.

The qualitative synthesis resulted in similar narratives among EMS workers, who felt they had gone through a traumatic experience, especially those working during the COVID-19 pandemic, and had to face various challenges (e.g., insufficient knowledge and training, poor PPE, constant protocol changes, concern regarding their own and their families' safety, etc.) but were willing to report for duty, as they understood and accepted the risks and felt a duty to perform their jobs. They did, however, feel there were acceptable limitations to this, such as having physical and mental health issues, lack of appropriate PPE, anti-viral medication, or appropriate vaccines, etc. In addition, participants expressed a strong need for reliable, trustworthy information, training, access to adequate and up-to-date PPE, proper communication from their organisations, and transparent protocols. The need for policies that are up-to-date, clear, and transparent was also highlighted by another recent publication [99] that investigated the public health regulations and policies dealing with preparedness and emergency management during the COVID-19 pandemic in Italy. They also concluded that more funds should be allocated in prevention, training, and information activities to make sure that we are better prepared for the next pandemic.

Participants stressed the importance of supportive relationships with significant others and their colleagues, as well as recognising that their organisations put effort into their comfort

and safety, which they valued and which made them feel safe. Others, though, felt tired and "left alone to fend" for themselves as they had to perform procedures outside of their job description and scope of practice or felt like they didn't receive enough support or information about getting tested (for COVID-19) nor any mental health or financial support from their employers. The importance of support (especially organisational support) in safeguarding frontline workers' mental health and well-being during the COVID-19 pandemic was also highlighted in a recent study [98] that explored the perceived support and pandemic preparedness among social workers in hospital settings in Israel. According to their results, only half of the social workers perceived receiving high levels of support during the COVID-19 pandemic; a finding echoing some of this review's results as well.

This review has also identified various gaps in current literature and the need for future research across various areas. None of the included studies reported on the experiences and views of patients that had been attended to by EMS during a disease outbreak/pandemic; this could be an area for future research to help us understand better the needs of this population. In addition, the majority of the reviewed studies were conducted in high-income countries, with predominantly White participants. More studies are, therefore, needed in low- and middle-income countries, with participants from diverse ethnic backgrounds, to evaluate whether the same interventions can be successfully implemented across different countries and populations. Further research with more diverse populations is also needed to explore potential factors affecting EMS providers' experiences, views, and attitudes towards working during pandemics. Equally, it would be crucial to investigate the long-term impact of working in a pandemic on the well-being and working conditions of emergency medical services and how these may be affected by the trajectory of the disease outbreak. Finally, the majority of the included studies did not report any or had missing data on the age, gender and/or ethnicity of their participants. Better reporting is, therefore, needed of demographic characteristics of the participants in published papers in the future.

## Strengths and limitations

This is the first scoping review of published studies that discuss EMS preparedness levels and aims to understand how the evidence translates into practice. This review has brought together papers discussing EMS preparedness during various disease outbreaks (including the current COVID-19 pandemic) and evaluating different types of interventions, as well as exploring EMS personnel's experiences of working during pandemics, and has synthesised them for the first time. The study followed a rigorous pre-specified protocol (registered with Open Science Framework), which ensured that the review process was transparent and replicable. We identified 90 studies for inclusion. The final development of themes (both quantitative and qualitative) was undertaken through discussion with the wider review team, consisting of reviewers from different backgrounds (e.g., medicine, nursing, and psychology).

This scoping review has some limitations. We are still learning to live with COVID-19 and more studies will be conducted and published as a result; therefore, the findings of this review are subject to change as more studies are being added to the existing literature base. Despite our efforts to be as inclusive as possible, studies and journal articles that have not been published yet, or are only available in languages other than English, have not been included in this review. Another limitation of this review was the absence of a subject librarian review of the search strategy, as this would have made for a stronger methodology. That said, many search terms were searched for in full text fields and there were further supplementary searches conducted to identify any additional references, which produced a high number of references to be screened. Finally, our synthesis of the 18 qualitative papers led to five main themes and

various sub-themes, but we did not conduct a detailed analysis, such as a content or thematic analysis. Future reviews could focus on, analyse and synthesise EMS personnel's experiences only and publish these as a systematic review and meta-synthesis study.

## Implications for policy and practice

Synthesising this literature has allowed us to explore EMS pandemic preparedness in various countries around the world, as well as to identify what interventions have been successfully implemented and to better understand the experiences of EMS staff during pandemics.

An important aspect of pandemic preparedness is making sure that EMS workers are able and willing to work during pandemics. Our findings have shown that the main factors affecting EMS personnel's willingness to report for duty are having adequate PPE and having access to vaccines for themselves and, ideally, their families as well. Other important factors were knowing and feeling prepared to perform their responsibilities (including having adequate knowledge and training for disease outbreaks), confidence about safety at work and trust in their employer. EMS organisations, therefore, should make sure they are appropriately equipped with up-to-date PPE and that they offer training on how to correctly use this equipment on a regular basis. Having the funds to procure enough vaccines and treatments, if available, for those who need it would also be essential for making EMS workers feel safer in case of a future pandemic. Participants also highlighted the importance of continuous training and education for pandemic preparedness. Therefore, detailed and frequent training on various aspects of infection control practices would also be advantageous, especially about basic knowledge of infectious diseases (including routes of transmission and signs/symptoms), patient and practitioner safety related to infectious and communicable diseases, and how to properly decontaminate and disinfect ambulances after each patient contact. Conducting regular pandemic exercises or having skill-based drills (similar to that of mass casualty and disaster drills) might also be helpful in increasing EMS personnel's knowledge levels and maintaining their skills, as this was suggested by the participants as well [89].

Participants also felt that having clear, transparent, and simple protocols was very important and that it helped them remain calm by following the instructions and guidelines, while at the same time knowing that their organisation is continuously reviewing and improving procedures made them feel safe and protected. Hence, EMS organisations should aim to develop agency/department-specific infection control policies and procedures (e.g., PPE requirements and the proper use of masks and respirators) that EMS workers can implement during routine calls, but also during a pandemic. Such initiatives would also increase trust in their employers, which as they stated was an important factor impacting their willingness to report for duty. Misinformation, lack of reliable data/information and communication about the pandemic from their employers was another one of the main concerns of the EMS workers; therefore, providing clear, up-to-date and accurate information (including detailed, localised data that could help them understand the geographic spread of cases throughout their area) on a regular basis would be imperative as well. This process could be facilitated by having a dedicated app or using social media, through which information could be delivered directly to EMS personnel, or even holding local question and answer sessions, as was suggested by the participants themselves [13]. Better communication and collaboration between departments and organisations, such as the emergency department and EMS, would also be helpful in ensuring that there is seamless continuation of care and less frustration for everyone involved.

Most participants also reported feeling stress, fear, anxiety, as well as feelings of frustration or failure, when they couldn't save a patient. Although opinions were divided and not everyone expressed a need for mental health support, offering such opportunities for employees who

need it would be important for their overall well-being. This is true especially given the fact that participants also reported not being able to socialise and take comfort in colleagues, while at the same time having less contact with their families, which they also found to be challenging. Peer support groups that can take place face-to-face or on-line, if needed to maintain social distancing rules, may be an ideal solution not only for the individuals (who have expressed the need for support from their team members and colleagues), but also for EMS organisations, as the financial cost would be minimal. Employers could also support EMS workers further by prioritising testing to keep them safe and to enable them to return to their duties, as well as by offering financial support in the case of illness, as participants also reported being unable to take any time off work in case of exposure and/or infection during the pandemic.

Many of the included papers in this review also implemented interventions that showed evidence of benefits in helping EMS services manage the COVID-19 pandemic and should be considered for further evaluation and adoption. These interventions included negative pressure ambulances and devices [28, 29, 47, 53]; a compact atmospheric plasma device [51]; and, performing chest compressions with the LUCAS 3 mechanical chest compression device [52]. In addition, having a dedicated COVID-19 EMS support track and/or video triage systems were found to be effective in handling the increased need for contact with EMS on call volume during the COVID-19 pandemic [54, 59, 61, 65] and similar triage measures could be applied to better manage future pandemics as well. Using discrete event simulation models [36] or first-stage optimization models [71], swarm intelligence algorithms [55, 56], Business Intelligence models [74] and/or the Internet of Things [76], could also help EMS to successfully manage the response to emergency calls.

Finally, our results showed that EMS staff are uniquely equipped to perform COVID-19 testing and participate in mass immunization efforts efficiently and safely. Therefore, EMS personnel can be a valuable resource in providing further services as well and help increase the number of persons that can be tested (even in more remote, underserved areas), while diminishing exposure to health care workers and other patients as well.

## Conclusions

Despite concerns for their own and their families' safety and the many challenges they were faced with, especially knowledge and training gaps, lack of appropriate PPE and constant protocol changes, EMS personnel were willing and prepared to report for duty during pandemics. Participants also made recommendations for future outbreak response, which should be taken into consideration in order for EMS to be better prepared to respond to any future pandemics.

## Supporting information

**S1 File. Preferred Reporting Items for Systematic reviews and Meta-Analyses extension for Scoping Reviews (PRISMA-ScR) checklist.**
(DOCX)

**S2 File. Ambulance (emergency medical service) interventions in response to pandemics: A scoping review protocol.**
(DOCX)

## Acknowledgments

We would like to thank Dr Withanage Iresha Udayangani Jayawickrama and Dr Sarathchandra Kumarawansa for their help conducting the initial searches for this scoping review. We

would also like to thank the members of the Community and Health Research Unit (CaHRU) study review group (University of Lincoln) for their valuable comments on a draft of this paper.

## Author Contributions

**Conceptualization:** Ffion Curtis, Aloysius Niroshan Siriwardena.

**Data curation:** Despina Laparidou.

**Formal analysis:** Despina Laparidou, Ffion Curtis, Nimali Wijegoonewardene, Joseph Akanuwe, Dedunu Dias Weligamage, Prasanna Dinesh Koggalage, Aloysius Niroshan Siriwardena.

**Investigation:** Despina Laparidou, Ffion Curtis, Nimali Wijegoonewardene, Joseph Akanuwe, Dedunu Dias Weligamage, Prasanna Dinesh Koggalage, Aloysius Niroshan Siriwardena.

**Methodology:** Despina Laparidou, Ffion Curtis, Aloysius Niroshan Siriwardena.

**Project administration:** Despina Laparidou.

**Resources:** Despina Laparidou.

**Software:** Despina Laparidou.

**Supervision:** Despina Laparidou, Ffion Curtis, Aloysius Niroshan Siriwardena.

**Validation:** Despina Laparidou, Ffion Curtis.

**Visualization:** Despina Laparidou.

**Writing – original draft:** Despina Laparidou.

**Writing – review & editing:** Despina Laparidou, Ffion Curtis, Nimali Wijegoonewardene, Joseph Akanuwe, Dedunu Dias Weligamage, Prasanna Dinesh Koggalage, Aloysius Niroshan Siriwardena.

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
