## [Decision Letter · Decision Letter 0]

11 Jan 2024

PONE-D-23-22267Emergency Medical Service interventions and experiences during pandemics: A scoping reviewPLOS ONE

Dear Dr. Siriwardena,

Thank you for submitting your manuscript to PLOS ONE. After careful consideration, we feel that it has merit but does not fully meet PLOS ONE’s publication criteria as it currently stands. Therefore, we invite you to submit a revised version of the manuscript that addresses the points raised during the review process.

We look forward to receiving your revised manuscript.

Kind regards,

Zeus Aranda, MSc MBA

Academic Editor

PLOS ONE

Journal Requirements:

3. We note that your Data Availability Statement is currently as follows: "All relevant data are within the manuscript and its Supporting Information files."

Reviewers' comments:

Reviewer's Responses to Questions

**Comments to the Author**

1. Is the manuscript technically sound, and do the data support the conclusions?

Reviewer #1: Yes

Reviewer #2: Yes

2. Has the statistical analysis been performed appropriately and rigorously? 

Reviewer #1: Yes

Reviewer #2: Yes

3. Have the authors made all data underlying the findings in their manuscript fully available?

Reviewer #1: Yes

Reviewer #2: Yes

4. Is the manuscript presented in an intelligible fashion and written in standard English?

Reviewer #1: Yes

Reviewer #2: Yes

5. Review Comments to the Author

Reviewer #1: Please see attached PDF for additional reviewer comments.

Regarding:overall recommendation "minor revision"  I think this is more along the lines of moderate, or between minor and major

1. Manuscript is technically sound although it does need some review

2. No major statistical analysis to speak of in this methodology

3. data contained in paper although I would encourage more clear search strategy reporting

4. Manuscript is overall intelligible although could do with significant review for clarity as many sections somewhat confusing requiring multiple read throughs.

Reviewer #2: Dear authors,

Although publications on COVID-19 have lost relevance after the wave of articles that "exhausted" and "fatigued" the scientific community, you have managed to present a work that collects evidence in a field little observed and of great relevance for care to emergencies and disasters (such as a pandemic); I congratulate you because the search has been rigorous and extensive, I see that it has faithfully concluded what other authors contributed from their studies and added a judicious and reasonable interpretation of the available evidence. Congratulations.

6. PLOS authors have the option to publish the peer review history of their article (what does this mean?). If published, this will include your full peer review and any attached files.

Reviewer #1: No

Reviewer #2: **Yes: **Luis Antonio Gorordo-Delsol

---

## [Author Response · Author response to Decision Letter 0]

29 Feb 2024

Dear Editor,

Please see the detailed response document for responses to reviewer comments and feedback.

1. Please ensure that your manuscript meets PLOS ONE style requirements, including those for file naming. We have adhered to the PLOS ONE requirements.

2. Data: All relevant data are within the manuscript and its Supporting Information files

3. We note that your Data Availability Statement is currently as follows: All relevant data are within the manuscript and its Supporting Information files. We confirm we do not have any raw data that are not included in the main and supporting papers. 

4. Please review your reference list to ensure that it is complete and correct. We have reviewed the reference list to ensure it is complete and correct. 

Yours faithfully,

 Professor A Niroshan Siriwardena MMedSci PhD FRCGP

---

## [Decision Letter · Decision Letter 1]

16 May 2024

Emergency Medical Service interventions and experiences during pandemics: A scoping review

PONE-D-23-22267R1

Dear Dr. Siriwardena,

We’re pleased to inform you that your manuscript has been judged scientifically suitable for publication and will be formally accepted for publication once it meets all outstanding technical requirements.

Kind regards,

Zeus Aranda, MSc MBA

Academic Editor

PLOS ONE

Additional Editor Comments (optional):

Reviewers' comments:

Reviewer's Responses to Questions

**Comments to the Author**

1. If the authors have adequately addressed your comments raised in a previous round of review and you feel that this manuscript is now acceptable for publication, you may indicate that here to bypass the “Comments to the Author” section, enter your conflict of interest statement in the “Confidential to Editor” section, and submit your "Accept" recommendation.

Reviewer #2: All comments have been addressed

2. Is the manuscript technically sound, and do the data support the conclusions?

Reviewer #2: Yes

3. Has the statistical analysis been performed appropriately and rigorously? 

Reviewer #2: Yes

4. Have the authors made all data underlying the findings in their manuscript fully available?

Reviewer #2: Yes

5. Is the manuscript presented in an intelligible fashion and written in standard English?

Reviewer #2: Yes

6. Review Comments to the Author

Reviewer #2: I appreciate the opportunity to review the manuscript again, I consider that the changes have been made, the work was enriched and it can be considered by the Editorial Committee for publication shortly.

7. PLOS authors have the option to publish the peer review history of their article (what does this mean?). If published, this will include your full peer review and any attached files.

Reviewer #2: **Yes: **Luis Antonio Gorordo-Delsol

---

## [Editor Report · Acceptance letter]

25 Jun 2024

PONE-D-23-22267R1 

PLOS ONE

Dear Dr. Siriwardena, 

I'm pleased to inform you that your manuscript has been deemed suitable for publication in PLOS ONE. Congratulations! Your manuscript is now being handed over to our production team.

Kind regards, 

on behalf of

Mr. Zeus Aranda 

Academic Editor

PLOS ONE